# Characterising the phenotypic evolution of circulating tumour cells during treatment

Simon Chang-Hao Tsao[1,2,3], Jing Wang[1],
Yuling Wang [1,6], Andreas Behren [2,4] Jonathan Cebon[2,3,4] & Matt Trau[1,5]

Real-time monitoring of cancer cells' phenotypic evolution during therapy can provide vital tumour biology information for treatment management. Circulating tumour cell (CTC) analysis has emerged as a useful monitoring tool, but its routine usage is restricted by either limited multiplexing capability or sensitivity. Here, we demonstrate the use of antibody-conjugated and Raman reporter-coated gold nanoparticles for simultaneous labelling and monitoring of multiple CTC surface markers (named as "cell signature"), without the need for isolating individual CTCs. We observe cell heterogeneity and phenotypic changes of melanoma cell lines during molecular targeted treatment. Furthermore, we follow the CTC signature changes of 10 stage-IV melanoma patients receiving immunological or molecular targeted therapies. Our technique maps the phenotypic evolution of patient CTCs sensitively and rapidly, and shows drug-resistant clones having different CTC signatures of potential clinical value. We believe our proposed method is of general interest in the CTC relevant research and translation fields.

[1] Centre for Personalised Nanomedicine, Australian Institute for Bioengineering and Nanotechnology, University of Queensland, Brisbane, QLD 4072, Australia. [2] Olivia Newton-John Cancer Research Institute, Heidelberg, VIC 3084, Australia. [3] Department of Surgery, University of Melbourne, Austin Health, Heidelberg, VIC 3084, Australia. [4] School of Cancer Medicine, La Trobe University, Bundoora, VIC 3086, Australia. [5] School of Chemistry and Molecular Biosciences, University of Queensland, Brisbane, QLD 4072, Australia. [6] Present address: Department of Molecular Sciences, Faculty of Science and Engineering, Macquarie University, Sydney 2109, Australia. These authors contributed equally: Simon Chang-Hao Tsao, Jing Wang. Correspondence and requests for materials should be addressed to Y.W. (email: yuling.wang@mq.edu.au) or to M.T. (email: m.trau@uq.edu.au)

The analysis of circulating tumour cells (CTCs) is emerging as a potentially valuable tool for monitoring cancer treatment response and understanding tumour biology from a simple blood test[1]. From a post-treatment clinical standpoint, it is important to determine (i) the impact of treatment on the disease, (ii) the presence of residual disease, (iii) the emergence of tumour cells that are treatment resistant, including tumour cells able to evade the immune system after immunotherapy, and (iv) the escape mechanisms, which will in turn allow the modification of the treatment approach. Therapeutic resistance may result from selective and/or adaptive pressure that encourages proliferation of the resistant cell population, which may be phenotypically distinct from their precursors in physical size, shape, and surface marker expression[1–4]. Thus, conventional CTC monitoring which targets precursor cells (e.g., by targeting the same surface markers) may fail to detect these vital phenotypically different resistant clones.

Presently, CTCs are first isolated prior to downstream phenotypic or geno-typic analysis[4]. Most antibody-dependent CTC isolation strategies rely on a single surface marker of interest, such as epithelial cell adhesion molecule (EpCAM). The CellSearch system, which is the only Food and Drug Administration (FDA)-approved CTC detection technology, is an example of such technique[4]. These strategies are prone to disregard tumour cells from (i) cancers of non-epithelial origin like melanoma, and (ii) cancers with downregulated EpCAM expression. The downregulation of EpCAM commonly occurs during epithelial-to-mesenchymal transition[1, 4], which is a process widely associated with treatment resistance in a variety of cancers[5]. On the other hand, antibody-free isolation strategies such as size-based separation often fail to isolate all relevant cells because of variable CTC physical properties[6, 7].

Following CTC isolation, downstream CTC phenotypic analysis mainly includes protein expression-based techniques such as flow cytometry, or nucleic acid-based techniques such as quantitative reverse transcription polymerase chain reaction (qRT-PCR)[4, 8]. Flow cytometry is one of the most commonly used techniques for cell characterisation but typically requires a relatively large quantity of sample cells and has limited multiplexing capabilities. New technologies such as CyTOF may be able to overcome these limitations;[9] however, it does not allow for the collection of live cells for further analysis or imaging afterwards. Although qRT-PCR is able to quantify relative expression of target transcripts within low quantities of CTCs, it is unable to directly quantify CTCs and determine their heterogeneity. Thus, an innovative method that allows direct phenotypic characterisation of multiple CTC surface markers with high sensitivity and without prior isolation is highly desired.

Here, we describe an approach for observing CTC phenotypic changes by monitoring the expression levels of multiple surface markers simultaneously via surface-enhanced Raman spectroscopy (SERS). SERS is a spectroscopic technique that possesses detection sensitivity down to single molecule level under certain conditions[10, 11] (such as when molecules are located in the "hot spots")[12, 13], and multiplexing capability[14, 15]. To demonstrate our technique, we test melanoma cell lines and melanoma CTCs, as melanoma is the deadliest form of skin cancer and has a rapid rise in incidence[16]. We select four melanoma CTC surface markers, including melanoma-chondroitin sulphate proteoglycan (MCSP)[17–22] and melanoma cell adhesion molecule (MCAM)[23–26] which are expressed in over 85 and 70% of the primary and metastatic melanoma lesions, respectively;[27, 28] erythroblastic leukaemia viral oncogene homologue 3 (ErbB3)[29], which is involved in therapy resistance development through activation of an alternative phosphoinositide 3-kinase–v-akt murine thymoma viral oncogene homologue (PI3K–AKT) pathway;[30, 31] and low-affinity nerve growth factor receptor (LNGFR)[32], a stem-cell biomarker which is strongly associated with resistance development[33]. The specific antibodies for targeting each surface marker are conjugated to SERS labels (i.e., Raman reporter-coated gold nanoparticles (AuNPs)), and a unique Raman spectrum (fingerprint) for each SERS label is generated upon a common laser wavelength excitation (Supplementary Fig. 1). The four Raman reporter-surface marker pairings are: 4-mercaptobenzoic acid (MBA) for MCSP; 2,3,5,6-tetrafluoro-4-mercaptobenzoic acid (TFMBA) for MCAM; 4-Mercapto-3-nitro benzoic acid (MNBA) for ErbB3; and 4-mercaptopyridine (MPY) for LNGFR (Supplementary Fig. 1). Detection specificity and sensitivity are assessed and validated using multiple cell lines and healthy donor samples. We then apply our methodology to monitor cellular phenotypic changes of melanoma cell lines harbouring BRAF mutations[34] in response to BRAF inhibitor (PLX4720). This newly FDA-approved drug could selectively inhibit mutated BRAF gene present in approximately 50% of melanoma[35]. We further examine blood samples collected serially from 10 stage-IV melanoma patients at different time points during their treatment course and monitor changes in their CTC phenotypes. We find that drug-resistant clones have different CTC phenotypes of potential clinical value.

## Results

**Working scheme**. The working principle of our method for phenotypic characterisation of CTCs from blood samples is illustrated in Fig. 1. Briefly, blood samples are processed for removal of red blood cells and leucocytes by density gradient centrifugation and CD45 depletion, respectively. Remaining cells are incubated with the four different antibody-conjugated SERS labels (Ab-SERS labels) and then simultaneously detected by Raman spectroscopy (Fig. 1a). Isotype-matched immunoglobulin (IgG)-SERS labels are used as an internal negative control in our experiments. For each sample, 150 measurements are continuously collected to represent different portions of cells that are undergoing Brownian motion in the solution (Fig. 1a). Each SERS measurement generates one SERS spectrum that is the statistically averaged result of a large ensemble of labelled cells within the scattering volume. The signal intensity is, therefore, proportional to the number of cells and their marker expression levels in the scattering volume.

The detection signals of each sample were analysed by profiling both surface marker signal distribution (Fig. 1b, c) and expression signatures (Fig. 1d). Spectral deconvolution was performed before signal analysis to minimise the potential overlap between characteristic peaks of different Raman reporters, in which a Gaussian function was used (Supplementary Fig. 2, and Supplementary Table 1). The signal distribution curve was then generated by plotting the Raman signal from each measurement (frequency vs. Raman intensity), thereby displaying the expression level distribution across all measured events. We hypothesised that the more diverse and heterogeneous the sample population, the wider the signal distribution of the respective markers (Fig. 1c). Hence, the selection of subclones or adaptation to specific selective pressure during treatment should result in a narrowed signal distribution spectrum to reflect more homogeneous phenotypes. In contrast, the signal distribution should broaden after resistance establishment. The cell signature, defined by the relative average expression levels of four markers, was extracted by collating the characteristic peak intensities of corresponding Ab-SERS labels with either MBA, TFMBA, MNBA, or MPY reporters (represented by peaks at 1075, 1375, 1335, and 1000 cm$^{-1}$, respectively). As cell populations with different phenotypes will have distinct combinations of surface marker expression levels, this profile is unique to each sample.

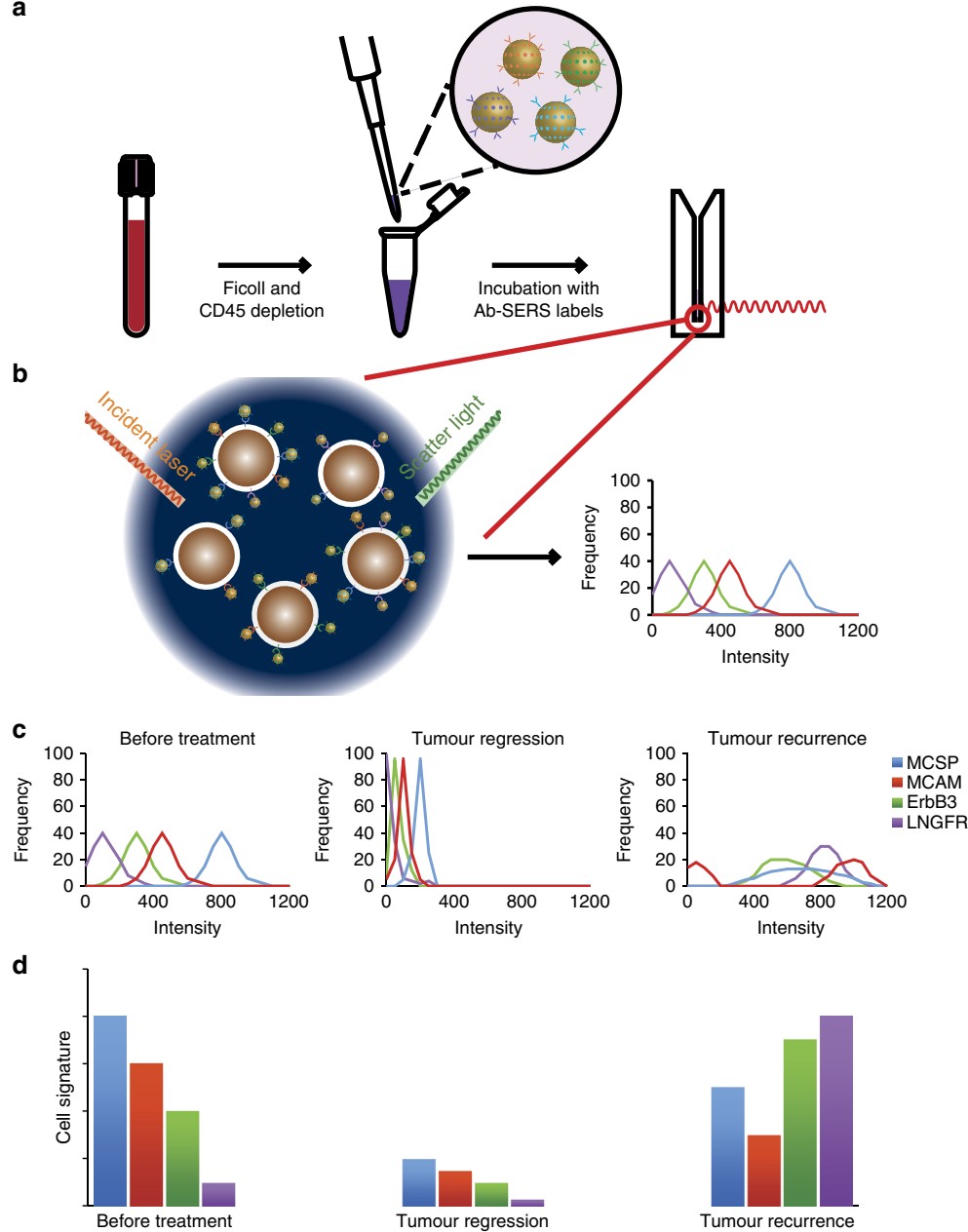

**Fig. 1** CTC detection and characterisation with Raman spectroscopy. **a**, **b** Schematics of experimental workflow: the blood sample taken from a patient is first depleted of RBC and PBMCs by processing over density gradient centrifugation (Ficoll) and subsequent CD45 depletion. Remaining cells are incubated with antibody-conjugated and Raman reporter-coated gold nanoparticles (Ab-SERS labels). The sample is subsequently washed and tested with Raman spectroscopy. To characterise the CTC populations, the Raman intensities are plotted as a frequency distribution curve. This curve represents the sample's range of expression. Higher intensity indicates the presence of more Ab-SERS labels as a result of higher marker expression levels or number of cells. Four melanoma surface marker antibodies (MCSP, MCAM, ErbB3, and LNGFR) with four specific SERS labels can be multiplexed for monitoring the CTC surface marker expression simultaneously; **c** CTC populations in response to treatment: the frequency distribution of each marker can signal how diverse the cell populations are in terms of surface marker expression levels. The more diverse and heterogeneous the sample population, the wider the signal distribution of the respective markers. Selection of subclones or adaptation to specific selective pressure results in on-treatment signatures with a narrowing spectrum of phenotypes, while after resistance establishment phenotypic spreading can be observed. **d** CTC signature in response to treatment: the relationship between the average Raman intensities of each surface maker represents the CTC signature. This signature is unique to each cell population. Shrinking of all marker intensities but retaining the relationships (CTC signature) could mean diminished cell number (tumour regression). Changing of CTC signature means the population now has a different phenotype, which could represent a treatment-resistant cell population

Single-cell SERS image was obtained by using the integrated Raman intensity of the characteristic peak from each Ab-SERS label. The colour (i.e., blue, red, green, and purple) of each spot in Fig. 2b represents the spatial distribution of Ab-SERS labels, which further indicates the cell surface marker distribution.

**Assay specificity**. To demonstrate the specificity of each Ab-SERS label alone and in combination, we first tested the performance of Ab-SERS labels in cell lines that have been well characterised and reported in literature[36–43]. Typically, SK-MEL-28, which has been reported for high expression of MCSP and MCAM[36], was chosen

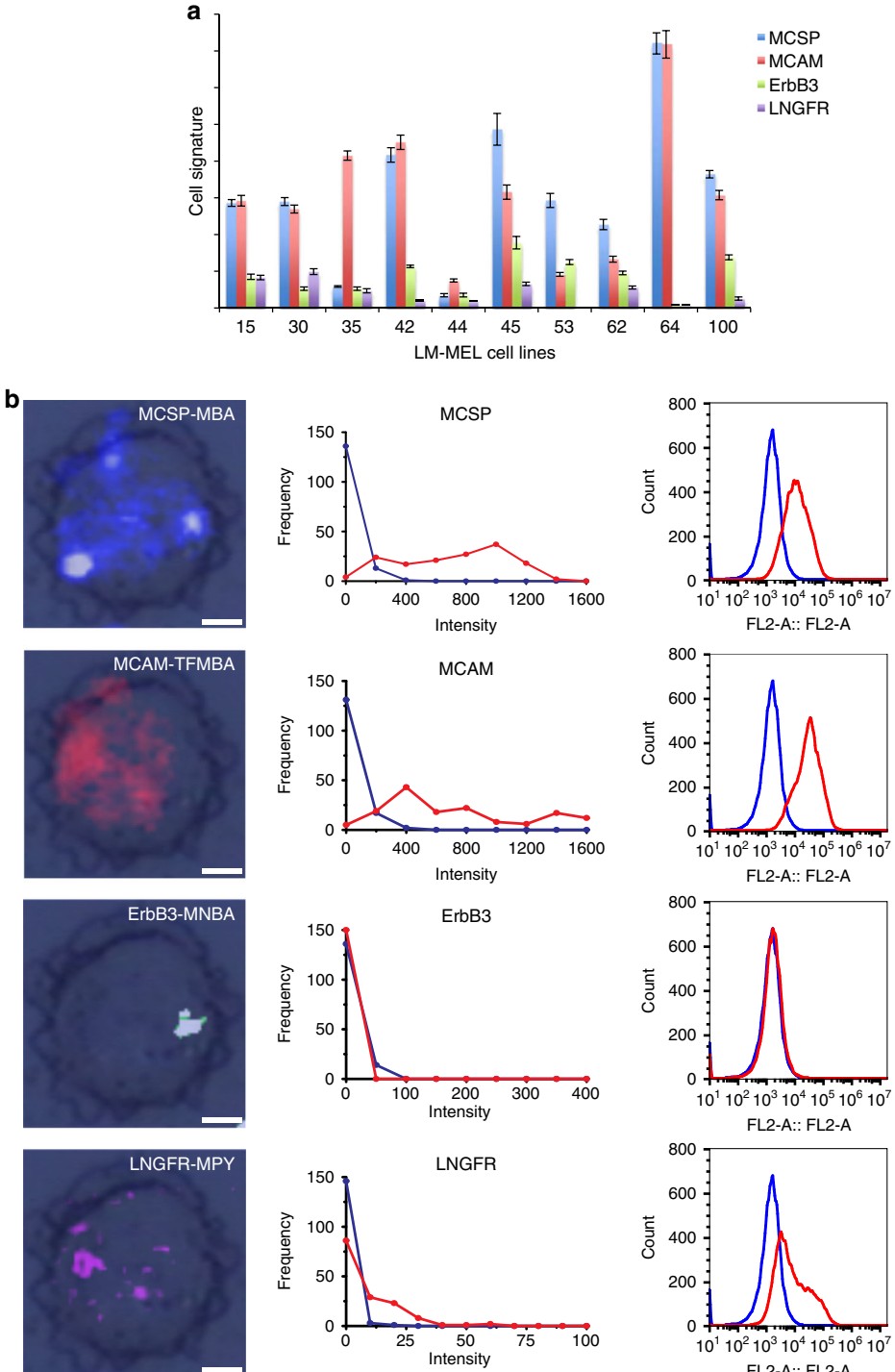

**Fig. 2** Cell signatures and surface marker expression profile. **a** Cell signatures of 10 melanoma cell lines. Melanoma cell lines' surface marker expression profiles identified by Raman spectroscopy. The unique pattern of cell surface marker expression (MCSP, MCAM, ErbB3, and LNGFR) as quantified by Raman spectroscopy, illustrating different phenotypes among different cell lines. **b** Surface marker expression profiles for LM-MEL-64. SERS images (left) of the distribution of four surface markers on a single cell and distribution obtained by Raman spectroscopy (middle). Flow cytometry (right) was used as a standard technique to characterise the surface marker expression of LM-MEL-64 cells. The result is comparable to the intensity distribution obtained by Raman spectroscopy (middle). In the presence of abundant cells, both techniques are able to separate marker-positive cells (red) from isotype control (blue). Data in **a** are mean ± s.d. Replicates are biological replicates ($n = 3$). Scale bars, 7 μm

as the marker-positive cell line for the specificity study of MCSP-SERS and MCAM-SERS labels. MCF7, which has low expression of MCSP and MCAM, was used as the marker-negative cell line[37–39]. The detailed information for all marker expression in

tested cell lines has been summarised in Supplementary Table 2[36–43]. Our data shown in Supplementary Fig. 3a and b were consistent to the literature reports[36–39], where we observed high SERS signals for MCSP-MBA and MCAM-TFMBA from

SK-MEL-28 cells, indicating high expression of MCSP and MCAM in SK-MEL-28 cells, and low SERS signals from MCF7, suggesting low expression of MCSP and MCAM in MCF7 cells. This result demonstrated the specificity of our SERS technique for the detection of MCSP and MCAM individually. Moreover, we tested the specificity of ErbB3 and LNGFR in MCF7 and SKBR3 cell lines, and in SK-MEL-28 and bone marrow mesenchymal stem cells, respectively. Different marker expression levels in these reported cell lines were successfully identified from resulting ErbB3-MNBA and LNGFR-MPY signals (Supplementary Fig. 3c, d), which were in line with literature reports[36, 40–43]. Furthermore, we tested the specificity of four antibodies together in both SK-MEL-28 and MCF7 cell lines (Supplementary Fig. 3e), which showed consistent results as determined by individual antibodies ($R^2 = 0.996$ for SK-MEL-28, and $R^2 = 0.985$ for MCF7, Supplementary Fig. 3f). To further demonstrate the specificity of our assay, flow cytometry detection was performed for validation and showed agreement with SERS data (Supplementary Fig. 4). Taken together, consistent results between SERS and flow cytometry data further demonstrated the high specificity of Ab-SERS labels along with minimum to no unspecific noise signal for cell surface marker detection.

To further validate that the proposed approach is capable of analysing CTC without an isolation step, our developed and well-characterised melanoma cell lines (LM-MEL-15, 30, 35, 42, 44, 45, 53, 62, 64, and 100)[34], one cervical cancer cell line (HeLa), and healthy donors' peripheral blood mononuclear cell (PBMC) samples were incubated with MCSP-SERS labels as proof-of-principle. MCSP is reported to be expressed on the majority of melanoma cell lines but absent on HeLa cells and PBMCs[36]. Our results showed varied MCSP expression across the different melanoma cell lines, while HeLa cells and PBMCs exhibited negligible background signals (Supplementary Fig. 5). SERS image of the mixed tumour cells and blood cells only indicated signals of MCSP-MBA-AuNPs from tumour cells, further confirming the high specificity of this approach (Supplementary Fig. 6).

**Assay sensitivity**. The sensitivity of using each Ab-SERS label alone and in combination for cell detection was explored by titrating 10–1000 cells (either SK-MEL-28 or MCF7) into 1 mL of PBS. As demonstrated in Supplementary Fig. 7, each antibody-SERS label alone and in combination enable the detection down to 10 cells, demonstrating that our technique is sensitive for cell characterisation.

To further evaluate the sensitivity of using our technique to detect CTCs, different numbers of our developed LM-MEL-64 cell line cells in 1 mL of PBS (Supplementary Fig. 8) and 10 mL of whole blood (Supplementary Fig. 9) were labelled with MCSP-SERS labels for detection. In Supplementary Fig. 8, Raman intensities showed a positive correlation with increasing cell numbers, and a detection limit of 10 cells was distinguished from the blank PBS signal. In Supplementary Fig. 9, although the average Raman intensities also increased with higher cell numbers, the results showed a relatively lower intensity and larger standard deviation, compared to those titrated into PBS (Supplementary Fig. 8). This is possibly due to the unpredictable loss of CTCs during sample processing, a well-known technical barrier. The signal with 250–1000 spiked cells was significantly lower as cells were lost more readily in those groups. Currently, a significant sensitivity improvement in patient samples has been achieved based on multi-molecular markers[25] or protein markers[18], compared to single marker assays. We thus used multiple markers to increase the probability of detecting the low quantity of remaining CTCs.

**Cell characterisation**. All four Ab-SERS labels were incubated with various melanoma cell lines to visualise their respective cell signatures. The distinct cell signatures among different cell lines were shown in Fig. 2a. For example, LM-MEL-64 has very high MCSP and MCAM expression levels but low ErbB3 and LNGFR levels, whereas LM-MEL-100 has lower MCSP and MCAM expression levels but significantly higher ErbB3 expression (Fig. 2a). Isotype controls were also performed and demonstrated that melanoma cells labelled with IgG-SERS labels exhibited negligible SERS signals (Supplementary Fig. 10). These data demonstrate that our method can distinguish tumour cells of differential surface marker expression signatures.

To demonstrate that four markers are useful for the detection of different CTC signatures, we applied linear discriminant analysis (LDA) to discriminate three typical melanoma cell lines using different numbers of markers as indicated in Supplementary Fig. 11. LDA is a statistical analysis that characterises or separates clusters based on the linear combination of features (i.e., cell signatures characterised by signals of target-specific SERS labels). We found that the discriminant function based on only one marker (i.e., MCSP) was unable to discriminate three melanoma cell lines, as shown in Supplementary Fig. 11a. With two markers (i.e., MCSP and MCAM), the discrimination accuracy for the three melanoma cell lines improved significantly (Supplementary Fig. 11b). Complete discrimination of three melanoma cell lines was achieved with discriminant functions generated by four markers (Supplementary Fig. 11c). Thus, these statistical data showed that these four markers were very helpful for the identification of melanoma cell subpopulations.

Figure 2b (left) shows SERS images of a single cell (LM-MEL-64) displaying each surface marker and demonstrating the multiplexing capability of Raman spectroscopy at single cell level. Figure 2b (middle) shows the Raman signal distribution constructed from 150 independent samplings (red) in comparison to isotype control (blue), exemplifying the heterogeneity within melanoma cell lines. The result matched those of flow cytometry (Fig. 2b (right)) using the same four antibodies but labelled with fluorophore-conjugated secondary antibodies instead of Ab-SERS labels. Five other cell lines (LM-MEL-15, 30, 35, 44, and 62) were also tested and correlated well with flow cytometry validation (Supplementary Figs. 12–16).

To directly visualise melanoma's cell heterogeneity within a single cell line, the distribution of four surface markers on each cell was imaged, and the relative expression level of each marker was compared (Supplementary Fig. 17). We investigated three cell lines (LM-MEL-33, 64, and 70) and five individual cells from each cell line. These data clearly showed cell heterogeneity with varied surface marker expression among individual cells of the same cell line. Hence, we believe that the highly sensitive and multiplexing capability of our proposed approach is ideal for characterising melanoma cells comprehensively.

**Cell line models in response to molecular targeted therapy**. To test the capability of our methodology in tracking the evolution of a resistant cell population, cellular phenotypic changes undergoing targeted therapy were assessed. Three melanoma cell lines harbouring an activating mutation in BRAF were treated continuously with PLX4720 (a BRAF inhibitor) to develop drug resistance. Surviving cells were obtained at regular intervals (days 0, 3, 7, 11, 17, 35, and 70). Within 3 days of drug treatment (day 3), distinct cell signatures were observed as compared to the respective controls (day 0, without drug treatment). Cell signatures then became stable after drug treatment for 11, 17, and 35 days (Fig. 3a), respectively. More importantly, these drug-treated melanoma cell lines displayed a similar cell signature after

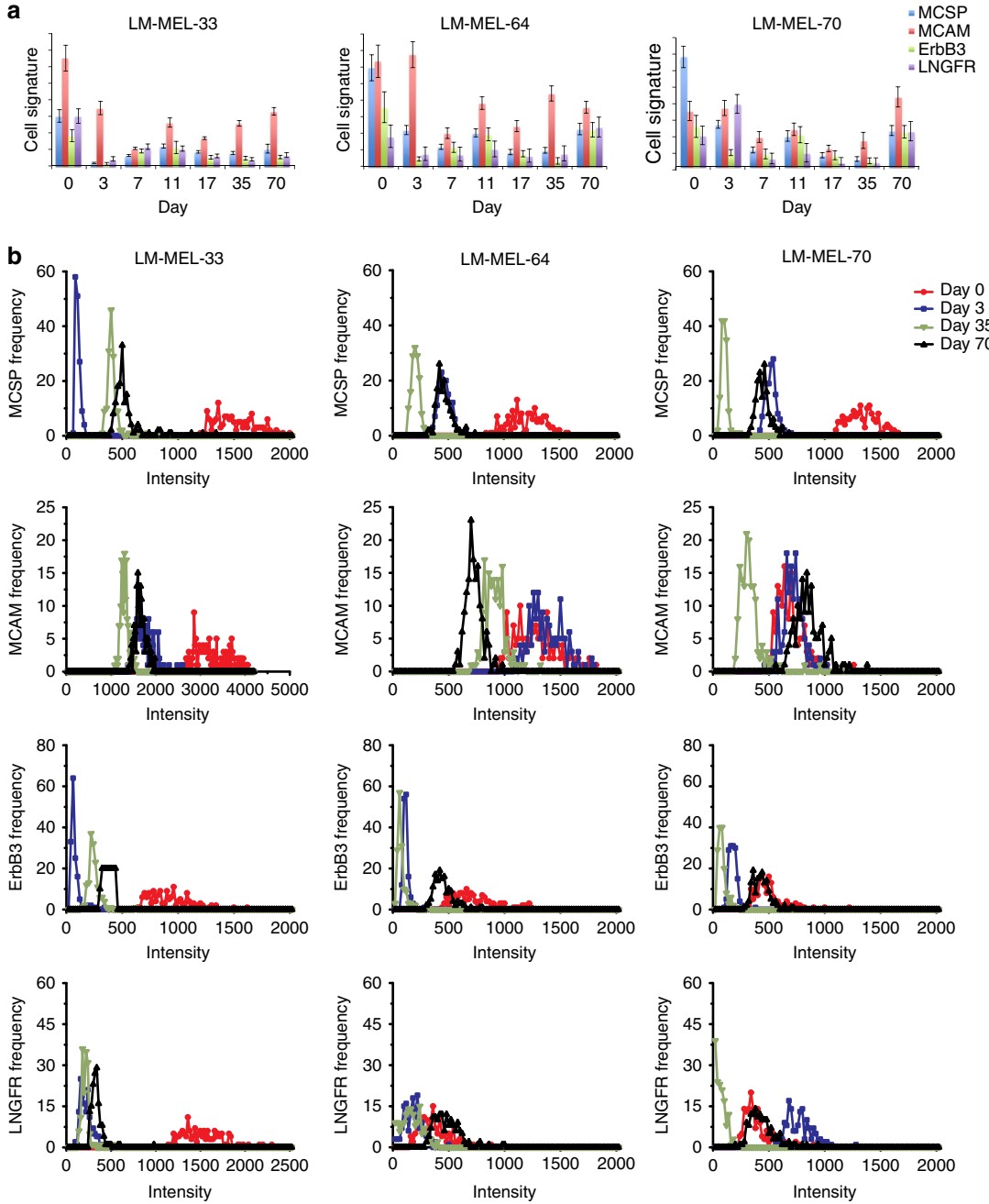

**Fig. 3** Cell phenotypes in response to drug treatment. **a** Cell signature and **b** distribution of cell surface marker expression levels before and during drug treatment (days 3, 35, and 70). Data in **a** are mean ± s.d. with 150 measurements. Each curve in **b** is calculated by 150 measurements of the bulk of cells

chronic PLX4720 exposure for 10 weeks with respect to the four markers measured, probably due to the effect of drug resistance selection. The signal distribution plots showed narrowed signal distribution (Fig. 3b) at early drug introduction, signifying drug selection of resistant clones and loss of the population heterogeneity. As the resistant clones expanded subsequently, we started to observe surface marker upregulation and signal distribution widening, thereby signifying proliferation and progression of the resistant clones. All of the cell line SERS data (Supplementary Figs. 18–21) have also been cross-validated with flow cytometry measurements (Supplementary Figs. 22–25), which displayed similar trends.

LDA was further applied to evaluate cell population shifts (based on SERS signals regarding four marker expression) in response to the drug treatment (Supplementary Fig. 26). For

visualisation of subpopulations, discriminant functions 1 and 2 derived from LDA were selected due to their relative efficiency in resolving cell line subpopulations. All three melanoma cell lines formed distinct subpopulations after drug treatment, and the subpopulations of drug-treated cell lines continuously shifted with drug treatment. This confirmed the effect of drug treatment on cell signatures, resulting in significantly different cell signatures from their parental counterparts.

**Patient CTCs in response to therapy.** To examine the capability of our method in monitoring patient therapy responses, we applied it to detect patient blood samples. Ten stage-IV melanoma patients' blood samples were serially collected during the course of treatment. Based on the initial radiological response to

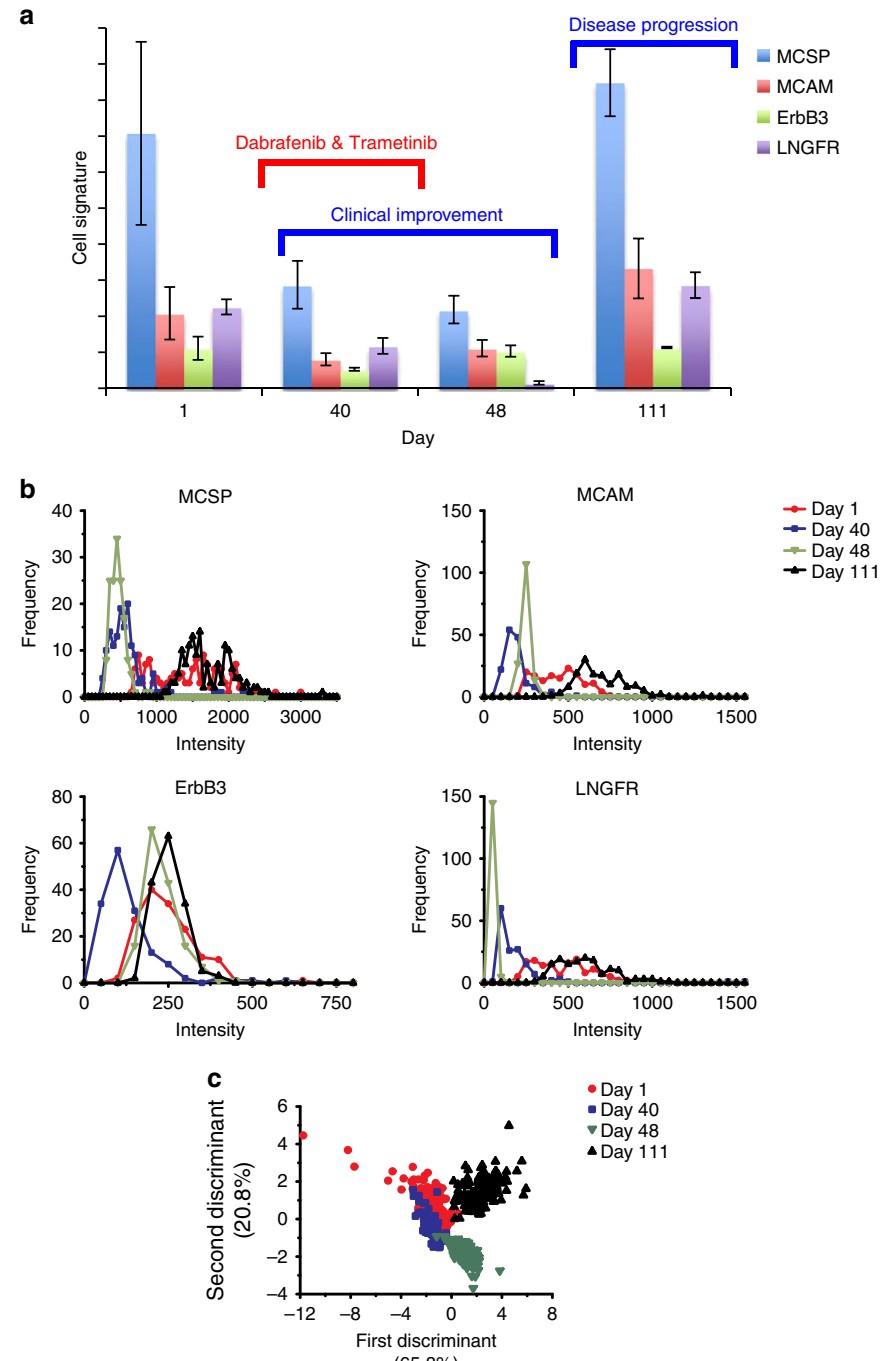

**Fig. 4** CTC signatures for patient 1. Patient 1 was treated with dabrafenib and trametinib for 1 month and discontinued because of toxicity. Tumour progressed after cessation of treatment. **a** CTC signatures presented according to days of treatment. All markers fell initially corresponding to clinical response (day 40). CTC signatures changed significantly on day 48 in response to treatment with markedly reduced LNGFR level. The signatures returned to pre-treatment pattern upon cessation of treatment, and elevated intensities correlated to disease progression (day 111). **b** Surface marker expression profile in response to treatment. Signal distribution indicated the tightening of the surface marker expression with treatment and broadening with disease progression. **c** Clustering of CTC signatures in response to therapy after application of LDA on SERS signals. Data in **a** are mean ± s.d. with 150 measurements. Each curve in **b** is calculated by 150 measurements of the bulk of cells

therapy, they could be broadly classified as: (1) treatment responders with objective diminish in tumour load, (2) non-responders with either increase or no radiological evidence of a decrease in tumour burden, and (3) mixed-responders, where some of the metastases responded, and others did not.

The CTC signatures and signal distribution for all patients during therapy treatment are shown in Fig. 4 and Supplementary Figs. 27–35. Changes to the CTC signatures showed variations in

expression levels of the respective markers (i) in relation to each other and (ii) overall in the CTC population. On the other hand, cell heterogeneity could often be seen in the treatment naïve or resistant group, which could be due to either plastic changes in gene expression or clonal selection of resistant cells.

For example, patient 1 (Fig. 4) received dabrafenib and trametinib, a BRAF-inhibitor and an MEK-inhibitor, respectively, for 1 month. Despite the objective clinical improvement,

treatment was discontinued because of toxicity. The CTC signature showed markedly reduced signal intensities in all markers with effective treatment, especially a significantly down-regulated LNGFR level (from 56% of the MCSP signal to just 4%) (Fig. 4a). However, as the patient's tumour progressed upon cessation of treatment, the CTC signature returned to pre-treatment pattern (with the return of LNGFR expression to 61% of the MCSP signal intensity). The signal distribution plot showed a much wider distribution during active disease and a significantly narrower distribution while the patient was on treatment (Fig. 4b). This could signify the selection pressure from treatment, resulting in the elimination of drug-sensitive cells. This pattern of "narrowing" signal distribution during treatment response was seen several times in our patient cohort. LDA was performed to statistically analyse all SERS signals that were collected from patient 1. Figure 4c shows that CTC populations shifted after drug treatment for 40 days and formed a total different cluster on day 48, indicating the shift of CTC populations in response to dabrafenib and trametinib treatment.

## Discussion

Current strategies for monitoring melanoma therapeutic resistance are insufficient. Radiological imaging (e.g., CT and PET) is insensitive to the detection of small lesions and provides limited information on tumour biology[44]. Although other many potential markers have been tested in melanoma, the only standard prognostic tumour marker—lactate dehydrogenase—is a non-specific enzyme that can be elevated in various benign or malignant diseases[45]. Circulating tumour DNA is a complementary marker to CTCs for detecting recurring disease and monitoring disease progression or therapeutic success;[4, 8] however, it cannot be used for phenotypic classification. Thus, CTC detection is anticipated to provide invaluable real-time biomarker information during treatment monitoring.

Compared with other CTC detection technologies[8], our study demonstrated an extremely sensitive, highly multiplexed, and simple method to rapidly detect real-time changes in CTC phenotypes and heterogeneity by monitoring surface protein expression profiles. We were able to detect 10 tumour cells in 10 mL of blood (Supplementary Fig. 9), which was comparable to other reported technologies such as the CellSearch system[46, 47] and CTC-Chip[48–51]. In addition, our method also displayed cell heterogeneity (Supplementary Fig. 17) and changes in tumour cell populations in response to molecular targeted therapy (Fig. 3). Currently, multi-parametric flow cytometry has been applied to detect multi-marker expressions in CTCs;[36] however, its detection sensitivity is ~2000–15,000-fold lower than SERS technology[52]. Finally, we have comprehensively profiled diverse CTC populations from 10 patient blood samples before treatment and at multiple time points during treatment (Fig. 4 and Supplementary Figs. 27–35), which clearly demonstrated the capability of our technology in a clinical setting. The high sensitivity of our technology could be attributed to three reasons: (1) no prior CTC isolation, that reduced CTC loss during the isolation process; (2) multi-marker-based CTC detection that increased the probability of detecting rare CTCs; (3) ultra-sensitive and multiplexed detection technology that allowed simultaneous characterisation of multiple markers expressed on the surface of rare CTCs. In comparison, antibody-dependent methods[21, 53], which require prior CTC isolation and/or rely on a single surface marker of interest, are prone to disregard tumour cells that have low target marker expression. Given that our technique is capable of effectively evaluating CTC phenotypes and heterogeneity in response to therapy, we thus believe it could be a big step towards understanding CTC characteristics (i.e., phenotypes) and promoting CTC clinical applications.

Sensitive and simple method to characterise CTC phenotypes in response to drug treatment could greatly improve our ability to study tumour's phenotypic alterations with treatment. This could help us understand important biological questions such as resistance mechanisms and discover novel therapeutic targets like receptor tyrosine kinases and other antibody targets. A recent report has demonstrated that subpopulations of melanoma CTCs show differential response to targeted therapy[36]. LNGFR has been described to be a potential marker of melanoma tumour stem cells with a high propensity to establish tumours[32, 54]. Other studies have also demonstrated that LNGFR is often upregulated and associated with resistance development[33, 55]. In line with these reports, patients 3 and 4 (Supplementary Figs. 28, 29) in our report both responded to immunotherapy (CheckMate trial) or targeted therapy (dabrafenib and trametinib) with significantly upregulated LNGFR expression on CTC surfaces. Concurrently, both patients' tumours developed resistance with subsequent worsening of disease after the last blood samples were taken (data not shown). Similarly, ErbB3 has been shown to be an important factor in resistance and metastasis development[29, 56], and can be seen to be upregulated in patient 3 and 6 who showed tumour progression while on treatment (Supplementary Figs. 28, 31).

In our study, we have also reiterated the importance of using multiple markers simultaneously in the detection of CTCs to increase sensitivity. Both patient 2 and 4 had markedly reduced MCSP expression post-treatment (Supplementary Figs. 27, 29). An isolation/detection technique targeting only MCSP would have failed to isolate the majority of their CTCs.

The number of melanoma CTCs has been shown to be prognostic of the overall survival in patients with metastatic melanoma[53]. It has been previously proposed that one could quantify CTC numbers by correlating the obtained Raman signal intensity to a cell number correlation curve generated using a cell line[57]. Our study also showed that the relative intensity change could provide a rough guide regarding to changes to CTC numbers within the same patient. It is worth noting that the observed signal changes may also arise from marker expression changes on a similar number CTCs. With simple modifications, such as a ferrous coating of our described NPs, our technique can potentially be used to capture and enumerate CTCs, and provide downstream analysis of other biomarkers such as epigenetic and transcriptional levels.

High multiplexing capability of Raman spectra (up to 31-plex)[14, 15] permits the incorporation of numerous markers to track CTCs' phenotypic changes with treatment. The technique has the potential to be adopted widely as it is becoming more affordable and portable (Supplementary Fig. 36 shows one typical SERS spectrum for one of the patient samples obtained from the hand-held Raman spectrometer). Healthcare institutions will be able to equip with such device for simpler treatment and disease monitoring.

## Methods

**Clinical sample acquisition**. This study was conducted according to the National Health & Medical Research Council Australian Code for the responsible conduct of Research and the National Statement on Ethical Conduct in Human Research. All patients have provided their written informed consent for the research study protocol, which was approved by the Human Research Ethics Committee of the Austin Hospital, Melbourne. Ethics approval was obtained from The University of Queensland Institutional Human Research Ethics Committee (Approval No. 2011001315). Methods pertaining to clinical samples were carried out in accordance with approved guidelines.

**Cell lines**. Twelve melanoma cell lines, LM-MEL-15, 30, 33, 35, 42, 44, 45, 53, 62, 64, 70, and 100, were established at the Ludwig Institute for Cancer Research in Melbourne and were authenticated by short-tandem repeat profiling. SK-MEL-28, SKBR3, MCF7, and HeLa cell lines were obtained from the American Type Culture Collection, being used for the specificity or sensitivity assay. Melanoma cells, SK-

MEL-28, SKBR3, MCF7, and HeLa cells, were maintained in RF10 medium which is made up of RPMI 1640 media (Invitrogen), 10% foetal calf serum (FCS) (CSL), 2 mM Glutamax (Gibco), and 1% PenStrep (Invitrogen). BD-MSC cell line was purchased from Rooster Bio company (Donor number: 0081) and cultured according to the standard method. BD-MSC at passage 13 was collected for the experiment. All cells were kept in a humidified incubator in 5% $CO_2$ at 37 °C. All cell lines were routinely tested for mycoplasma.

**Treatment with BRAF inhibitor.** Melanoma cells (LM-MEL 33, 64, and 70) were grown in medium with 1 μM PLX4720 (Selleckchem).

**PBMC isolation.** Blood samples were collected in EDTA containing 50 mL falcon tubes and processed within 4 h from collection over Ficoll-Paque PLUS (GE Healthcare Life Science), according to manufacturer's protocol. Isolated PBMC from each 10 mL of blood were stored in a CryoTubes (Corning) containing 80% RF10, 10% DMSO, and 10% FCS at −80 °C. Healthy donors' blood was obtained from Red Cross blood bank and processed the same way as patient samples.

**CD45 depletion.** PBMCs were depleted with the EasySep Human CD45 depletion kit (StemCell), according to the manufacturer's protocol.

**Flow cytometry.** Flow cytometry was performed on BD Accuri™ C6. Cells resuspended in 200 μL of FACS buffer (PBS containing 3% FCS, 1% BSA, and 1 mM EDTA) were incubated with 0.25 μg of either anti-MCSP, anti-MCAM, anti-ErbB3, or anti-LNGFR mouse monoclonal antibodies (MAB2585, MAB932, MAB348/MAB3481, and MAB367, R&D Systems) or isotype-matched control (Normal mouse IgG sc-2025, Santa Cruz Biotech) prior to staining with labelled secondary antibodies (Alexa Fluor 488 goat anti-mouse IgG antibody, A-11001, Life Technologies) diluted 1:2000 in FACS buffer. Data were analysed with BD Accuri™ C6 software.

**Ab-SERS label preparation.** Gold(III) chloride trihydrate ($HAuCl_4 \cdot 3H_2O$), MBA, 5, 5′-dithiobis (2-nitrobenzoic acid) (DTNB), TFMBA, MPY, sodium borohydride ($NaBH_4$), 11-mercaptoundecanoic acid (MUA), bovine serum albumin (BSA), 4-(2-hydroxyethyl)-1-piperazineethanesulfonic acid (HEPES), N-(3-dimethyl-aminopropyl)-N′-ethyl-carbodiimide (EDC), and sulfo-N-hydroxy-sulfosuccinimide (Sulfo-NHS) were purchased from Sigma, Aldrich, Fluka, Thermo Scientific, respectively. Tris-sodium citrate ($Na_3$-citrate) was bought from Ajax Finechem. To synthesise MNBA, fresh 300 μL of 20 mM $NaBH_4$ was added into 2 mL of 5 mM DTNB to break the disulphide bond in DTNB through reducing reaction.

Ab-SERS labels were prepared by functionalizing AuNPs with Raman reporters and antibodies. Briefly, 60 nm AuNPs were synthesised by citrate reduction of $HAuCl_4$[58]. 100 mL of $HAuCl_4$ ($10^{-2}$% by weight) was heated to boiling, and 0.7 mL of $Na_3$-citrate (1% by weight) was then added. The mixture was continously boiling for 20 min and then cooled down to room temperature (RT) for further functionalization. 10 μL of 1 mM Raman reporters (MBA, MNBA, MPY, or TFMBA) and 2 μL of 1.0 mM MUA (antibody conjugation linker) in ethanol were then added into 1 mL of AuNP suspension. The mixture was incubated for 5 h at RT to form a complete self-assembled monolayer. The mixture was then centrifuged at 7600 rpm for 10 min to remove residual reactants and resuspended in 200 μL of HEPES buffer (pH = 5.9). Afterwards, carboxyl groups of MUA were activated by EDC and Sulfo-NHS (40 μL of 3.33 mg mL$^{-1}$ EDC and 40 μL of 2 mg mL$^{-1}$ Sulfo-NHS) in HEPES buffer at RT for 20 min under shaking. SERS labels were then centrifuged to remove excess EDC and Sulfo-NHS and redispersed into 200 μL of 0.1 mM PBS. 1 μg of either human anti-MCSP, anti-MCAM, anti-ErbB3, and anti-LNGFR mouse monoclonal antibodies (MAB2585, MAB932, MAB348/MAB3481, and MAB367, R&D Systems) or isotype-matched IgG (Normal mouse IgG sc-2025, Santa Cruz Biotech) was then added to the mixture and incubated for 0.5 h at RT. After that, Ab-SERS labels were centrifuged at $600 \times g$ at 4 °C for 8 min to remove free antibodies and resuspended in 200 μL of 0.1% BSA for 0.5 h at RT to block non-specific binding sites. To minimise the settlement effect of large Ab-SERS labels, Ab-SERS labels were centrifuged at $400 \times g$ for 2 min before being applied for labelling.

**Ab-SERS labelling for cell line and CTC detection.** Cells suspended in 200 μL of buffer (PBS containing 1% FCS) were incubated with the mixture of four Ab-SERS labels (30 μL each) at 37 °C for 30 min followed by gentle centrifuge at $400 \times g$ for 1 min and washing with 200 μL of buffer. The washing step was repeated for four times. The samples were then re-suspended in 60 μL of buffer and placed into a cuvette for SERS measurements. To minimise data variations caused by different cell numbers, the same amount of cells from different cell lines were tested across the assays.

Patient sample experiments were done in a blinded fashion. All patient samples had been de-identified by a research assistant not involved in the experiment and only revealed after the spectra were analysed. All samples were prepared and measured on the same day at the same time using the same batch of SERS labels.

**SERS measurement.** SERS spectra were recorded with a portable IM-52 Raman Microscope (Snowy Range Instruments). The 785 nm laser wavelength was used

for excitation of Raman scattering. SERS spectra were obtained at 1 s integration time with a laser power of 70 mW. SERS images were recorded with the Witec alpha 300 R microscope with 632.8-nm line from a HeNe laser as excitation and obtained at 100 ms integration time with an EMCCD, using a 20× microcopy objective.

**SERS spectral analysis.** A Gaussian function was applied to deconvolute the resulting Raman spectrum into separate sources to minimise the potential overlap between characteristic peaks, using Fityk 0.9.8 program[59]. With this function, we fitted peaks or separated close peaks according to peak positions, intensities, and full width at half maximum.

**Statistical analysis.** LDA was performed with SPSS 19.0 software package (SPSS Inc., Chicago, IL).

**Data availability.** The data that support the findings of this study are available from the corresponding author on request.

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

## Acknowledgements

This work was supported by the ARC DECRA (DE140101056) awarded to Dr. Y.W. We would like to acknowledge the funding support from the National Breast Cancer Foundation of Australia to M.T. (CG-12-07). These grants have significantly contributed to the environment to stimulate the research described here. The Raman imaging was conducted at Queensland node of the Australian National Fabrication Facility (Q-ANFF). S.C.T. is supported by the Australian Postgraduate Award and the Royal Australian College of Surgeons Foundation for Surgery Research Scholarship. J.W. is supported by the Australian Government Research Training Program Scholarships. A.B. is supported by a MCRF (17019) from the Victorian Cancer Agency (VCA). We also acknowledge Kevin M. Koo and Dr. Darren Korbie for manuscript revision. We would also like to acknowledge Joanne Hawking and Christopher Hudson for their assistance with patient sample collection. We would also like to thank Ocean Optics for generously loaning the handheld Raman spectrometer used in one of patient sample tests. Most importantly, we would like to thank all the patients for their generosity and support.

## Author contributions

S.C.T., J.W., Y.W., A.B., and M.T. designed the research and analysed the data. S.C.T., J. W., and Y.W. performed the experiments and prepared the manuscript. A.B., J.C., and M.T. commented on the manuscript.

## Additional information

**Competing interests:** The authors declare no competing interests.

