## [Peer Review File(PDF 5607 kb) · Nature Communications]

Reviewers' comments:

Reviewer #1 (Remarks to the Author):

While the article demonstrates a novel approach to analysing the phenotypic evolution of circulating tumour cells during treatment, I do not believe the paper is ready to be accepted into a publication such as Nature Communications. This research could be of potential importance to scientists in this field and of great interest to those in related disciplines and so I would advise the following suggestions and comments are taken into account before a resubmission is considered. The main weaknesses of the paper are the quality of writing and the clarity of presented data. The quality of writing throughout the article must be addressed. Specific notes are given below with accompanying line number, however it should be stressed that the text should be as succinct and direct as possible. At points in the review, phrases are repeated unnecessarily and communicated ideas could be made more brief. A dialogue about current methods for CTC phenotypic evaluation would be a worthwhile addition, as well as a brief discussion on the literature surrounding treatment resistance so that the importance of this research is emphasised in the correct context. Regarding the clarity of presented data, specific points are also given below. While presenting the information in the form of sample signal distribution may be specifically useful for this spread of data, the explanation of the significance of what the data represents could be delivered with more clarity below Figure 1 and in the main text for better access to non-specialists. For example, in Figure 1c, the sample signal distribution for tumour recurrence has not been discussed. Also, the inclusion of the isotope control was a necessity but its use needs to be highlighted in the main text.

Additional Comments

Lines 16-17: Build upon the idea of treatment resistance, background info, cite literature where this has been studied.

Line 19: Reconsider use of word 'hampered'.

Lines 36-40: Would not agree that these are most pressing issues from clinical standpoint on cancer patient, but rather that they are the most pressing issues for a cancer patient post-treatment. Also, consider rewording of this sentence, maintain clarity and be succinct.

Line 38: 'residual'

Lines 40-41: 'selective or adaptive pressures' are referenced but no further explanation given. If a huge impact of this research is identifying phenotypic changes, a more in depth but brief background of phenotypic changes should be included.

Lines 47-50: As EpCAM mediated CTC capture and EMT is a crucial example of phenotypic change which obstructs CTC isolation, perhaps these concepts shouldn't be presented in brackets as a side note but more of a central illustration.

Line 55: Existing methods of CTC isolation have been highlighted in this paragraph but still no background as to current approaches used for phenotypic characterisation, which is perhaps more pertinent to this paper.

Line 58: 'Multiple surface protein expressions' singular, not 'proteins'

Line 61: Reference to 100 markers detected simultaneous via single laser, cannot find reference to this in the papers cited. Double check this figure as it seems to be inflated.

Line 62: 'clinical samples' perhaps instead of 'patient bloods'

Line 64: Perhaps include references to validate the surface markers chosen.

Line 68: 'Ra-AuNP' As Ra is element radium, perhaps consider alternative abbreviation.

Line 68: Specific dye used for each corresponding antibody should be named.

Line 67-68: Consider rewording, sounds as if antibodies conjugated... as a SERS label. Need to separate the ideas.

Line 72: 'CD45 depletion, respectively'

Line 73: 'antibody conjugated'

Line 75: Repeat of 'single cell level'

Line 76: Consider rewording use of 'interrogated'.

Line 80: It is stated that the average signal intensities are collated but not specified at which point

the intensities have been measured.

Line 93: Again, reference to 100 marker simultaneous detection.

Line 97: Stated '10 well characterised melanoma cell lines' but have listed only 8.

Line 108: '10-1000 melanoma cells into 1 ml of PBS' - If cells have already been incubated with MCSP-reporter-AuNPs at this point, must refer to them in conjugate form and not just as cells.

Clarity of methodology is key.

Line 119: 'Sample processing' not procession.

Line 129: 'This demonstrates that our method...'

Line 130: Here, consider stating that the method has the potential to detect changes in patient CTC phenotypes as up to this point in paper you have only demonstrated method on spiked samples.

Line 213: 'which can then be subsequently analysed downstream.'

Line 215: Consider rewording sentence, also another reference to the 100 markers.

Line 221: Consider rewording sentence beginning 'More hospitals...'

Figure 1- (a/b) Incident and scattered light could be labelled as such (c) the resulting sample signal distribution for tumour recurrence should be explained with more clarity in the main text.

Figure 2- (a) The colour used to represent MCAM has changed from the previous data in Fig 1 (orange to red). Keep all presented data sets consistent throughout. (b) Consider re-ordering the presentation of data, with SERS images then Raman intensity distribution and lastly flow cytometry on the right.

Supplementary Information

Reference to EDC-NHS chemistry has been made but it has not been clarified as to whether a PEG linker has been used. Figures 4 + 5 - State below figures the cell line analysed and which surface marker has been targeted.

Reviewer #2 (Remarks to the Author):

This research proposes the possibility of real-time monitoring of circulating melanoma cells (CTCs). The technology uses antibodies conjugated with Raman reporter coated gold nanoparticles and uses these as labels for surface enhanced Raman scattering labels to simultaneously monitor heterogeneous CTCs. Authors suggest that the technology can be used to measure changes to the CTC population during treatment and possible use this information to select therapeutic options for patients.

This is a potentially powerful technology but several of the suggested claims made by the authors need additional support. The following changes are suggested.

1. Melanoma CTCs are extremely heterogeneous. Only four antibodies conjugated with Raman reporter coated gold nanoparticles are used for this study. Claims are made about this being useful for greater than 100. It needs to be demonstrated that this technology can be used for more than four. A rationale was provided for the selection of these markers but there are others that could have been more useful especially related to resistance. Why were these markers excluded?
2. The authors need to provide data as to the exact number of antibodies conjugated with Raman reporter coated gold nanoparticles that would be useful for CTC population shifts in the clinic. Is four enough or what would the proper number be to clinically useful? Statistically justification would be useful.
3. No comparison to currently used technologies was provided. This comparison should at least have made in the discussion. Since most current CTC isolation technologies (Cell Search) use 7 ml of blood and detect sometimes as few as 2-5 CTC; would this technology be useful clinically? How useful would this technology be on a 7 ml sample containing 5 heterogeneous CTCs? Also, is this approach superior to circulating tumor DNA technologies?
4. This technology is restricted to the cell surface. However, most accepted diagnostic, prognostic or therapeutic markers for melanoma are internal to the cell. It would be useful if this technology

could also be used for measurement of internal CTC markers. Also, this article stresses resistance development but none of the markers selected were cell surface markers used in drug resistance development. This seems like a serious deficiency?

5. The lack of an animal model makes most of the claims only correlative. An animal model using spontaneously metastasizing CTC would assist in supporting many of the correlative claims made in the article. This would be especially important for the claims of being able to track evolution of a resistant CTC population. Mixing resistant and nonresistant cell lines would be ideal to demonstrate the utility of this technology. These models are widely available. Authors are strongly encouraged to add these models as it would significantly improve the impact of the work.

6. Manuscript seems to lack statistical analysis.

Detailed Responses to Reviewer's Comments

Reviewer 1

1. While the article demonstrates a novel approach to analysing the phenotypic evolution of circulating tumour cells during treatment, I do not believe the paper is ready to be accepted into a publication such as Nature Communications. This research could be of potential importance to scientists in this field and of great interest to those in related disciplines and so I would advise the following suggestions and comments are taken into account before a resubmission is considered.

Reply: We appreciate the Reviewer's recognition of the importance of this work and the valuable comments. All comments are addressed below.

2. The main weaknesses of the paper are the quality of writing and the clarity of presented data. The quality of writing throughout the article must be addressed. Specific notes are given below with accompanying line number, however it should be stressed that the text should be as succinct and direct as possible. At points in the review, phrases are repeated unnecessarily and communicated ideas could be made more brief.

Reply: We appreciate the Reviewer's comments on the writing and the clarity of presented data. We have now thoroughly revised the writing and data accordingly to improve the manuscript quality.

3. A dialogue about current methods for CTC phenotypic evaluation would be a worthwhile addition, as well as a brief discussion on the literature surrounding treatment resistance so that the importance of this research is emphasised in the correct context.

Reply: We agree with the Reviewer on enhancing the literature review and discussion for the current methods for CTC study and treatment resistance. We now have added two paragraphs in the manuscript:

“Following CTC isolation, downstream CTC phenotypic analysis mainly includes protein expression-based techniques such as flow cytometry; or nucleic acid-based techniques such as quantitative reverse transcription polymerase chain reaction (qRT-PCR) (Krebs, M.G., *et al. Nat. Rev. Clin. Oncol.* 2014; Alix-Panabieres, C., *et al. Nat. Rev. Cancer* 2014; Alix-Panabieres, C., *et al. FEBS let.* 2017). Flow cytometry is one of the most commonly used techniques for cell characterisation but typically requires a relatively large quantity of sample cells and has limited multiplexing capabilities. New technologies such as CyTOF may be able to overcome these limitations (Leipold, M. D. *et al. JoVE* 2012), however it does not allow for collection of live cells for further analysis or imaging afterwards. Although qRT-PCR is able to quantify relative expressions of target transcripts within low quantities of CTCs, it is unable to directly quantify CTCs and determine their heterogeneity. Thus, an innovative method that allows for direct phenotypic characterisation of multiple CTC surface markers with high sensitivity and without prior isolation is highly desired.”

This paragraph is included in our revised manuscript (Pages 3-4).

“Current strategies for monitoring melanoma therapeutic resistance are insufficient. Radiological imaging (*e.g.* CT and PET) is insensitive in detecting small lesions and only able to provide limited information on tumour biology (O'Connor, J. P.; *et al. Clin. Cancer Res.* 2015). Although other correlate markers have been tested in melanoma, the only standard prognostic tumour marker - lactate dehydrogenase (LDH) - is a nonspecific enzyme that can be elevated in various benign or malignant diseases (Díaz-Lagares, A. *et al. Tumor Biology*

2011). Circulating tumour DNA (ctDNA) is a complementary biomarker to CTCs for detecting recurring disease and monitoring disease progression or therapeutic success (Krebs, M. G., *et al. Nat. Rev. Clin. Oncol.* 2014; Alix-Panabieres, C. *et al. Nat. Rev. Cancer* 2014); however it is not able to be designed for phenotypic classification nor the isolation of CTCs. Thus, CTC detection is anticipated to provide invaluable biomarkers for real-time treatment monitoring.”

This paragraph is included in the “Discussion” section of our revised version (Page 10).

4. Regarding the clarity of presented data, specific points are also given below. While presenting the information in the form of sample signal distribution may be specifically useful for this spread of data, the explanation of the significance of what the data represents could be delivered with more clarity below Figure 1 and in the main text for better access to non-specialists. For example, in Figure 1c, the sample signal distribution for tumour recurrence has not been discussed. Also, the inclusion of the isotope control was a necessity but its use needs to be highlighted in the main text.

Reply:

a. According to the reviewer’s comment, we have now modified the legend of Fig. 1 (Page 17) and the main text (Page 5), in particular Fig. 1c, for the sample signal distributions for tumour recurrence.

We reworded the legend of **Fig. 1c** as:

“(c) The more diverse and heterogeneous the sample population, the wider the signal distribution of the respective markers. Selection of subclones or adaptation to specific selective pressure results in on-treatment signatures with a narrowing spectrum of phenotypes, while after resistance establishment phenotypic spreading can be observed.”

In the main text, we added additional sentences to clarify the resulting sample signal distribution for tumour recurrence:

“The signal distribution curve was generated by plotting the Raman signal from each measurement (frequency versus Raman intensity), thereby displaying the expression level distribution across all measured events. We propose that the more diverse and heterogeneous the sample population, the wider the signal distribution of the respective markers (**Fig. 1c**). Hence, selection of subclones or adaptation to specific selective pressure during treatment should result in a narrowed signal distribution spectrum to reflect more homogeneous phenotypes. In contrast, signal distribution should broaden after resistance establishment.”

b. The isotope control has also been highlighted in the revised manuscript (Pages 5&7).

Additional comments:

1. Lines 16-17: Build upon the idea of treatment resistance, background info, cite literature where this has been studied.

Reply: This information is now included in the introduction (**Page 3**).

2. Line 19: Reconsider use of word ‘hampered’.

Reply: We have changed “hampered” to “restricted”.

3. Lines 36-40: Would not agree that these are most pressing issues from clinical standpoint on cancer patient, but rather that they are the most pressing issues for a cancer patient post-treatment. Also, consider rewording of this sentence, maintain clarity and be succinct.

Reply: We reworded the sentence in the revised manuscript (**Page 3**) to “From a post-treatment clinical standpoint, it is important to determine (i) the impact treatment has on the disease, (ii) the presence of residual disease, (iii) the emergence of tumour cells that are treatment resistant, including tumor cells able to evade the immune system after immunotherapy, and (iv) the escape mechanisms which in turn will allow the modification of the treatment approach.”

4. Line 38: ‘residual’

Reply: We have now corrected the error.

5. Lines 40-41: ‘selective or adaptive pressures’ are referenced but no further explanation given. If a huge impact of this research is identifying phenotypic changes, a more in depth but brief background of phenotypic changes should be included.

Reply: We have now included additional information in the revised manuscript (**Page 3**):

“Therapeutic resistance may result from selective and/or adaptive pressure that encourages proliferation of the resistant cell population, which may be phenotypically distinct from their precursors in physical size, shape and surface marker expression (Diaz, L. A., *et al. Nature* 2012; Aparicio, S., *et al. N Engl J Med* 2013; Joosse, S. A., *et al. EMBO Mol Med* 2015; Alix-Panabières, *et al. Nat. Rev. Cancer.* 2014). Thus, conventional CTC monitoring which targets precursor cells (e.g., by targeting the same surface markers) may fail to detect these vital phenotypically-different resistant clones.”

6. Lines 47-50: As EpCAM mediated CTC capture and EMT is a crucial example of phenotypic change which obstructs CTC isolation, perhaps these concepts shouldn’t be presented in brackets as a side note but more of a central illustration.

Reply: We modified the sentence in **Page 3** to:

“Most antibody-dependent CTC isolation strategies rely on a single surface marker of interest, such as epithelial cell adhesion molecule (EpCAM). The CellSearch system, which is the only Food and Drug Administration (FDA)-approved CTC detection technology, is an example of such technique (Alix-Panabieres, C., *et al. Nat. Rev. Cancer* 2014). These strategies are prone to disregard tumour cells from (i) cancers of non-epithelial origin like melanoma, and (ii) cancers with down-regulated EpCAM expression. The down-regulation of EpCAM commonly occurs during epithelial-to-mesenchymal transition (EMT), which is a process widely associated with treatment resistance in a variety of cancers.”

7. Line 55: Existing methods of CTC isolation have been highlighted in this paragraph but still no background as to current approaches used for phenotypic characterisation, which is perhaps more pertinent to this paper.

Reply: Current approaches used for phenotypic characterisation have been reviewed and added in the revised manuscript (**Pages 3-4**).

8. Line 58: ‘Multiple surface protein expressions’ singular, not ‘proteins’

Reply: We have now corrected the error.

9. Line 61: Reference to 100 markers detected simultaneous via single laser, cannot find reference to this in the papers cited. Double check this figure as it seems to be inflated.

Reply: As reported in the literature by Bernat Mir-Simon *et al. (Chem. Mater.* 2015) and Sebastian Schlücker (*ChemPhysChem* 2009), we made a claim in the original manuscript that the SERS technology can *theoretically* be multiplexed for greater than 100 markers because of the strikingly unique and easily discernible Raman spectra (fingerprints) created by different molecular reporters on the surface of gold nanoparticles. As described by Sebastian Schlücker

(*ChemPhysChem* 2009), in theory, there are unlimited chemicals that could generate unique Raman spectra and thus discernible Raman fingerprints/signatures/barcodes. However, so far, only 31 chemicals have been experimentally reported as Raman reporters used for multiplexing (Mir-Simon, B., *et al. Chem. Mater.* 2015). Hence, taking into account that the 100 figure was a theoretical value, not yet achieved experimentally, and taking into account the reviewer's comments, we have revised the manuscript and added two new references (**Page 4**) to include a more conservative maximum number of 31 for the scalability of the multiplexing.

10. Line 62: 'clinical samples' perhaps instead of 'patient bloods'

Reply: We have now corrected the error.

11. Line 64: Perhaps include references to validate the surface markers chosen.

Reply: Additional references were cited to validate the surface markers chosen and included in the revised manuscript (**Page 4**).

12. Line 68: 'Ra-AuNP' As Ra is element radium, perhaps consider alternative abbreviation.

Reply: We corrected 'Ra-AuNP' to 'SERS labels'.

13. Line 68: Specific dye used for each corresponding antibody should be named.

Reply: We have modified this in the manuscript (**Page 4**):

The four Raman reporter-surface marker pairings are: 4-mercaptobenzoic acid (MBA) for MCSP; 2,3,5,6-tetrafluoro-4-mercaptobenzoic acid (TFMBA) for MCAM; 4-Mercapto-3-nitro benzoic acid (MNBA) for ErbB3; and 4-mercaptopyridine (MPY) for LNGFR (**Supplementary Fig. 1**).

14. Line 67-68: Consider rewording, sounds as if antibodies conjugated... as a SERS label. Need to separate the ideas.

Reply: We reworded the sentence in the manuscript (**Page 4**) to:

"The specific antibodies for binding each surface marker target are conjugated to SERS labels (i.e., Raman reporters-coated gold nanoparticles), and a unique Raman spectrum (fingerprint) for each marker target is generated upon a common laser wavelength excitation (**Supplementary Fig. 1**)."

15. Line 72: 'CD45 depletion, respectively'

Reply: We have now corrected the error.

16. Line 73: 'antibody conjugated'

Reply: We have now corrected the error.

17. Line 75: Repeat of 'single cell level'

Reply: We deleted the sentence "...which can ultra-sensitively interrogate samples at a single cell level."

18. Line 76: Consider rewording use of 'interrogated'.

Reply: We changed to "acquired".

19. Line 80: It is stated that the average signal intensities are collated but not specified at which point the intensities have been measured.

Reply: The cell signature, defined by the relative average expression levels of four markers, was extracted by collating the characteristic peak intensities of corresponding Ab-SERS labels

with either MBA, TFMBA, MNBA, or MPY reporters (represented by peaks at 1075 cm⁻¹, 1375 cm⁻¹, 1335 cm⁻¹, and 1000 cm⁻¹, respectively).

This point has also been clarified in our revised version (Page 5).

20. Line 93: Again, reference to 100 markers simultaneous detection.

Reply: We added the references in revised manuscript (Page 4).

21. Line 97: Stated ‘10 well characterised melanoma cell lines’ but have listed only 8.

Reply: We added “45” & “53” in revised manuscript (Page 6).

22. Line 108: ‘10-1000 melanoma cells into 1 ml of PBS’ - If cells have already been incubated with MCSP-reporter-AuNPs at this point, must refer to them in conjugate form and not just as cells. Clarity of methodology is key.

Reply: We corrected to “The sensitivity of our approach was determined by titrating 10 to 1000 LM-MEL-64 cells into 1 mL of PBS (Supplementary Fig. 4) and 10 mL of whole blood (Supplementary Fig. 5), respectively. MCSP-MBA-AuNPs were then added for cell labelling.” in the revised manuscript (Page 6).

23. Line 119: ‘Sample processing’ not procession.

Reply: We have now corrected the error.

24. Line 129: ‘This demonstrates that our method...’

Reply: We have now corrected the error.

25. Line 130: Here, consider stating that the method has the potential to detect changes in patient CTC phenotypes as up to this point in paper you have only demonstrated method on spiked samples.

Reply: We have now corrected the error in the revised manuscript (Page 7).

26. Line 213: ‘which can then be subsequently analysed downstream.’

Reply: We have now corrected the error.

27. Line 215: Consider rewording sentence, also another reference to the 100 markers.

Reply: We reworded the sentence to “High multiplexing capability of Raman spectra (up to 31-plex) permits the incorporation of numerous markers, to track CTCs’ phenotypic changes with treatment.”

References (Mir-Simon, B., *et al. Chem. Mater.* 2015; Schlücker, S., *Chemphyschem* 2009) were cited in the revised manuscript (Page 12).

28. Line 221: Consider rewording sentence beginning ‘More hospitals...’

Reply: We reworded to “Healthcare institutions will be able to equip with such device for easy treatment and disease monitoring.” in the revised manuscript (Page 12).

29. Figure 1- (a/b) Incident and scattered light could be labelled as such (c) the resulting sample signal distribution for tumour recurrence should be explained with more clarity in the main text.

Reply: We have labelled the incident and scatter light in Figure 1-(a/b).

Figure 1-c: the resulting sample signal distribution for tumour recurrence has now been explained with more clarity in the legend of Fig. 1c (Page 17) and the main text (Page 5).

30. Figure 2- (a) The colour used to represent MCAM has changed from the previous data in Fig 1 (orange to red). Keep all presented data sets consistent throughout. (b) Consider re-ordering the presentation of data, with SERS images then Raman intensity distribution and lastly flow cytometry on the right.

Reply: Figure 2-(a): the colour has been changed to keep consistent across the whole work; (b) we have reordering the presentation of data according to Reviewer's comment, with SERS images then Raman intensity distribution and lastly flow cytometry on the right.

31. Supplementary Information: Reference to EDC-NHS chemistry has been made but it has not been clarified as to whether a PEG linker has been used.

Reply: PEG linker was not used in our study, instead, 11-mercapto-1-undecanoic acid (MUA) was used as the linker for antibody conjugation. This point has been clarified in our revised supplementary (Page 5).

32. Supplementary Figures 4 + 5 – State below figures the cell line analysed and which surface marker has been targeted.

Reply: The detailed information for cell lines and the target surface biomarker have been stated in Supplementary Figures 4+5 in the revised Supplementary Document.

Reviewer #2

This research proposes the possibility of real-time monitoring of circulating melanoma cells (CTCs). The technology uses antibodies conjugated with Raman reporter coated gold nanoparticles and uses these as labels for surface enhanced Raman scattering labels to simultaneously monitor heterogeneous CTCs. Authors suggest that the technology can be used to measure changes to the CTC population during treatment and possible use this information to select therapeutic options for patients.

This is a potentially powerful technology but several of the suggested claims made by the authors need additional support. The following changes are suggested.

Reply: We appreciate the Reviewer's recognition of the potential powerful technology proposed in this work and the valuable comments. All comments are addressed below.

1. Melanoma CTCs are extremely heterogeneous. Only four antibodies conjugated with Raman reporter coated gold nanoparticles are used for this study. Claims are made about this being useful for greater than 100. It needs to be demonstrated that this technology can be used for more than four. A rationale was provided for the selection of these markers but there are others that could have been more useful especially related to resistance. Why were these markers excluded?

Reply:

(a) We agree with the Reviewer that melanoma CTCs are extremely heterogeneous. Thus, our foremost goal was to demonstrate the ability of our novel technology to study key aspects of cell heterogeneity quantitatively (e.g., detailed and multiplexed surface receptor distribution profiles) on a relatively simple platform that has sufficient detection sensitivity for clinical analysis. Hence, as an initial technology demonstration, we selected a panel of four well-known melanoma surface markers. We have shown that these are indeed already able to characterise cell heterogeneity (**Figure R1**) and track phenotypic shifts in melanoma CTCs under treatment in the clinic – despite their limited number. A key beauty of the SERS nanoparticle system is that the multiplexing is highly scalable, and promising biomarkers with available antibodies could readily be further incorporated for clinical benefit in the future.

(b) As reported in the literature by Bernat Mir-Simon *et al.* (*Chem. Mater.* 2015) and Sebastian Schlücker (*ChemPhysChem* 2009), we made a claim in the original manuscript that the SERS technology can *theoretically* be multiplexed for greater than 100 markers because of the strikingly unique and easily discernible Raman spectra (fingerprints) created by different molecular reporters on the surface of gold nanoparticles. As described by Sebastian Schlücker (*ChemPhysChem* 2009), in theory, there are unlimited chemicals that could generate unique Raman spectra and thus discernible Raman fingerprints/signatures/barcodes. However, so far, only 31 chemicals have been experimentally reported as Raman reporters used for multiplexing (Mir-Simon, B., *et al. Chem. Mater.* 2015). Hence, taking into account that the 100 figure was a theoretical value, not yet achieved experimentally, and taking into account the reviewer's comments, we have revised the manuscript (**Page 4**) to include a more conservative maximum number of 31 for the scalability of the multiplexing. Such a high number would of course also lead to potential crowding of the SERS nanoparticles on the surface of the cells, and so the maximum number would more than likely be somewhat less than this also. Our recent study on cancer molecular subtyping (Koo, K.M., *et al. Small*, 2016) has also demonstrated the potential of this technology for simultaneously detecting 5 biomarkers. We thus firmly believe that the SERS multiplexing capability is highly scalable as the reporter technology improves.

(c) There is not a common surface marker (or even an established panel) that would cover the possibilities of resistance acquisition for BRAF inhibition, neither for immunotherapy

currently (Luke, J. J., *et al. Nat. Rev. Clin. Oncol.* 2017). Thus, what we demonstrate here is the potential to look for those once a consensus panel of markers (extra and intracellular) has been established. Our technology is highly adaptable and could be flexibly modified for the detection of other promising biomarkers now or in the future. Additionally, expression levels of the selected four biomarkers are shown through our experiments to vary and associate with drug therapy. Hence, we believe these four biomarkers may have potential clinical value.

(d) Furthermore, we have now also carried out a significant amount of single cell imaging experiments, where we have used the 4 already described antibody-SERS labels as multiplexed optical labels that have been imaged directly using a Raman Microscope to visualise the distribution of these particles on single cells (Figure R1). The new data has now also been included in the revised manuscript (Supplementary Fig. 13). With the four biomarkers labelled with antibody-SERS labels, the distribution of four biomarkers on the single cell has been imaged and the relative expression level of each biomarker has been compared. This new data clearly shows melanoma cell heterogeneity that surface biomarker expression varies among individual cells. Hence, we believe that the highly sensitive and multiplexing capability of our proposed approach is ideal for characterising melanoma cells comprehensively.

This point has been added in our revised version (Page 8).

2. The authors need to provide data as to the exact number of antibodies conjugated with Raman reporter coated gold nanoparticles that would be useful for CTC population shifts in the clinic. Is four enough or what would the proper number be to clinically useful? Statistically justification would be useful.

Reply:

To demonstrate that the four biomarkers are useful for the detection of different CTC signatures, we applied linear discriminant analysis (LDA) to discriminate three typical melanoma cell lines using different numbers of biomarkers as indicated in Figure R2. LDA is a statistical analysis that characterises or separates clusters based on the linear combination of features (i.e., cell signatures characterised by signals of target-specific SERS nanoparticles). We found that the discriminant function based on only one biomarker (i.e., MCSP) was unable to discriminate three melanoma cell lines, as shown in Figure R2a. With two biomarkers (i.e., MCSP and MCAM) the discrimination accuracy for the three melanoma cell lines improved significantly (Figure R2b). Complete discrimination of three melanoma cell lines was achieved with discriminant functions generated by four biomarkers (Figure R2c). Thus, these statistical data show that these four biomarkers were very helpful for the identification of melanoma cell subpopulations. Furthermore, LDA results indicate that these four biomarkers are also able to track phenotypic shifts in melanoma cell lines in response to BRAF inhibition (Figure R3) and in CTCs under treatment in the clinic (Figure R4).

Moreover, the multiplexing capability of SERS is scalable, allowing for higher multiplexed detection to meet the clinical requirement.

This has been added in our revised version (Pages 7, 8 & 10).

3. No comparison to currently used technologies was provided. This comparison should at least have made in the discussion. Since most current CTC isolation technologies (Cell Search) use 7 ml of blood and detect sometimes as few as 2-5 CTC; would this technology be useful clinically? How useful would this technology be on a 7 ml sample containing 5 heterogeneous CTCs? Also, is this approach superior to circulating tumour DNA technologies?

Reply:

(a) We are grateful for this comment. We believe our technology is useful and powerful for clinical applications: (i) we easily detected 10 tumour cells in 10 mL of blood (**Supplementary Fig. 5**), which was comparable to other reported technologies such as the CellSearch system (Allard, W. J. *et al. Clin. Cancer Res.* 2004; Riethdorf, S., *et al. Clin. Cancer Res.* 2007) and CTC-Chip (Nagrath, S. *et al. Nature* 2007; Talasz, A. H. *et al. Proc. Natl Acad. Sci. USA*, 2009; Ozkumur, E. *et al. Sci. Transl. Med.* 2013; Saliba, A. E. *et al. Proc. Natl Acad. Sci. USA*. 2010; Krebs, M.G. *et al. Nat. Rev. Clin. Oncol.* 2014; Poudineh, M. *et al. Nat. Nanotechnol* 2017). (ii) In addition to CTC detection, our method also displayed cell heterogeneity (**Figure R1**), which is unachievable for the CellSearch system that possesses limited multiplexing capability (e.g., melanoma CTCs are identified by positive expression of MCSP) (Khoja, L. *et al. J. Invest. Dermatol.* 2013; Pantel, K. *et al. Nat. Rev. Cancer* 2008; Krebs, M.G. *et al. Nat. Rev. Clin. Oncol.* 2014; Alix-Panabieres, C. *et al. Nat. Rev. Cancer* 2014). (iii) We comprehensively profiled diverse CTC populations from 10 patient blood samples before treatment and at multiple time points during treatment (**Fig. 4 and Supplementary Figs. 17-25**), which clearly demonstrated the capability of our technology in a clinical setting.

This point has been added in the discussion (**Page 10**) as requested by the reviewer.

(b) In regards to CTC and ctDNA technologies, as CTC and ctDNA are complementary biomarkers used for real-time monitoring of therapeutic efficacy, both technologies have benefits in the clinic. However, ctDNA is not able to be designed for phenotypic classification nor the isolation of CTCs. Our CTC methodology is ideal for CTC surface biomarker characterisation and could be extended for an in-depth assessment of CTCs at various levels (DNA, RNA, and proteins). Our CTC analysis also allows the visualization of intact CTCs for morphological identification of a malignant phenotype. None of these is possible through ctDNA analysis alone.

This point has been addressed in the discussion (**Page 10**) according to Reviewer's comment.

4. This technology is restricted to the cell surface. However, most accepted diagnostic, prognostic or therapeutic markers for melanoma are internal to the cell. It would be useful if this technology could also be used for measurement of internal CTC markers. Also, this article stresses resistance development but none of the markers selected were cell surface markers used in drug resistance development. This seems like a serious deficiency?

Reply:

(a) This technology is not limited to the cell surface markers as there are many ways that particles can be engineered for intercellular study through cell-lysis, uptake of SERS nanoparticles into cells etc (Kang, J. W. *et al. Nano Lett.* 2015; Koo, K.M., *et al. Small*, 2016). However, as our first proof of concept and highly focused study on this technology, direct detection of cell-surface markers is more preferable considering the following advantages: (i) simple procedures which minimize possible loss of CTCs during sample treatment; (ii) preservation of intact cells for morphological identification.

(b) In regards to the selection of cell surface markers for monitoring drug resistance development, LNGFR and ErbB3 are recognised as important markers during the development of drug resistance in melanoma. They have been shown to often be upregulated (Landsberg, J. *et al. Nature* 2012; Fallahi Sichani, M. *et al. Mol Syst Biol* 2017; Abel, E. V. *et al. J. Clin. Invest.* 2013; Tiwary, S. *et al. Oncology* 2014). Whilst the aim of our study is not to investigate the specific resistance mechanism within treated patient populations, our selected markers serve as an example of biologically relevant molecules that are expressed differently during

treatment and resistance development. We have now added a paragraph in **Page 11** to stress this fact:

“LNGFR has been described to be a potential marker of melanoma tumour stem cells with a high propensity to establish tumours (Boiko, A. D.; *et al. Nature* 2010; Civenni, G.; *et al. Cancer Res.* 2011). Other studies have also demonstrated that LNGFR is often upregulated and potential assisting resistance development (Landsberg, J. *et al. Nature* 2012; Fallahi Sichani, M. *et al. Mol Syst Biol* 2017). In line with these reports, patient 3 and 4 (**Supplementary Figs. 18 and 19**) in our report both responded to immune (CheckMate trial) or molecular targeted therapy (dabrafenib and trametinib) with significantly upregulated LNGFR expression on CTC surfaces. Concurrently, both patients’ tumours developed resistance with subsequent worsening of disease after the last blood samples were taken (data not shown). Similarly, ErbB3 has been shown to be an important factor in resistance and metastasis development (Abel, E. V. *et al. J. Clin. Invest.* 2013; Tiwary, S. *et al. Oncology* 2014), and can be seen to be upregulated in patient 3 and 6 who showed tumour progression whilst on treatment (**Supplementary Figs. 18 and 21**).”

5. The lack of an animal model makes most of the claims only correlative. An animal models using spontaneously metastasizing CTC would assist in supporting many of the correlative claims made in the article. This would be especially important for the claims of being able to track evolution of a resistant CTC population. Mixing resistant and nonresistant cell lines would be ideal to demonstrate the utility of this technology. These models are widely available. Authors are strongly encouraged to add this models as it would significantly improve the impact of the work.

Reply:

(a) We thank the reviewer for this suggestion. We have now performed (and have included into the manuscript) new biological model data to support the claims in the manuscript. In order to track the evolution of a resistant CTC population, we have now included melanoma cell line experimental data in which we monitored cell phenotypic changes in response to drug treatment. The data presented in **Figure R5**, represent roughly two years’ worth of experiments (originally intended for a follow on paper) which clearly demonstrate similar trends in the model cell line data that we see within the patient longitudinal samples. All of the cell line data has also been cross-correlated with flow cytometry measurements (**Figure R6 and Figure R7**), which also detect similar trends.

We found that a decrease in both cell heterogeneity and expression levels after the initiation of drug treatment and a gradual recovery over time, demonstrating the ability of our technology to track the phenotypic changes during resistance development. This data has been discussed and added as **Figure 3** in the revised manuscript (**Page 8**).

“To test the capability of our methodology in tracking the evolution of a resistant CTC population, cellular phenotypic changes undergoing targeted therapy were assessed. Three melanoma cell lines harbouring an activating mutation in BRAF were treated continuously with PLX4720 (a BRAF inhibitor) to develop drug resistance. Surviving cells were obtained at regular intervals (Days 0, 3, 7, 11, 17, 35 and 70). Within 3 days of drug treatment (Day 3), distinct cell signatures were observed as compared to the respective controls (Day 0, without drug treatment). Cell signatures then became stable after drug treatment for 11, 17, and 35 days (**Fig. 3a**), respectively. More importantly, these drug-treated melanoma cell lines displayed a similar cell signature after chronic PLX4720 exposure for 10 weeks with respect to the four markers measured, thus indicating the effect of drug resistance selection. The signal distribution plots show narrowed signal distribution (**Fig. 3b**) at early drug introduction,

signifying drug selection of resistant clones and loss of the population heterogeneity. As the resistant clones expanded subsequently, we started to observe surface marker up-regulation and signal distribution widening, thereby signifying proliferation and progression of the resistant clones. All of the cell line data have also been cross-validated with flow cytometry measurements (**Figure R6 and Figure R7**), which also displayed similar trends.

LDA was further applied to evaluate cell population shifts (based on SERS signals regarding to four biomarker expressions) in response to the drug treatment (**Figure R3**). For visualization of subpopulations, discriminant functions 1 and 2 derived from LDA were selected due to their relative efficiency in resolving cell line subpopulations. All three melanoma cell lines form distinct subpopulations after drug treatment and the subpopulations of drug-treated cell lines continuously shift with drug treatment. This confirms the effect of drug treatment on cell signatures, resulting in significantly different cell signatures from their parental counterparts.”

(b) With regards to the suggestion for animal model data, given the clear trends we observed in the cell line model data (now included in our manuscript **Page 8**), which generally correlate with what we have observed in 10 longitudinal patient studies (each with numerous pre- and post-therapy time point analyses), we believe that the inclusion of animal model studies would not be of additional supportive value to our present data. Firstly, syngeneic animal models that allow for real-time measurement of treatment response to immunotherapy are very good tools, but may not reflect the same phenotypic shifts or melanoma surface markers like human patient samples. Secondly, animal studies can often display artefacts that are not evident in human systems (different pharmacokinetics etc). Thirdly, we have already moved to study significant numbers of patient samples in a longitudinal setting (in addition to comprehensive cell line data that shows similar general trends).

6. Manuscript seems to lack statistical analysis.

Reply: Thanks for your suggestions.

We have now applied linear discriminant analysis (LDA) to statistically analyse cell line models and CTCs from clinical samples. The analyses encompass the discrimination (based on SERS signals regarding to four surface marker expressions) of various melanoma cell lines (**Figure R2**); and evaluation of cell line and CTC surface marker expression in response to drug treatment (**Figures R3 & R4**). For the evaluation of cell line models with drug treatment (**Figure R3**), discriminant functions 1 and 2 derived from LDA were selected for visualization of subpopulations, considering their relative efficiency in resolving cell line subpopulations. It can be found that all three melanoma cell lines form distinct subpopulations after drug treatment and the subpopulations of drug-treated cell lines continuously shift with drug treatment. This confirms the effect of drug treatment on cell signatures, which results in significantly different cell signatures to their parental counterparts. For the evaluation of CTCs from patients with drug treatment, **Figure R4** shows CTC populations shifted after drug treatment for 40 days and formed a total different cluster on day 48, indicating the shift of CTC populations in response to dabrafenib and trametinib treatment.

These data have been added as **Fig. 4c and Supplementary Figs. 7&16** and discussed in the revised manuscript (**Pages 7, 8 and 10**).

LM-MEL-33

LM-MEL-64

Figure R1. Single-cell SERS images of LM-MEL-33, LM-MEL-64, and LM-MEL-70. The y-axis (percentage) in histogram is calculated by counting the numbers of dots that generating positive signals on the individual cell (i.e. Percentage = the pixel numbers of dots/the pixel numbers of per cell). Blue, red, orange, and green represent MCSP, ErbB3, LNGFR, and MCAM, respectively.

Figure R2. Clustering three melanoma cell lines (LM-MEL-33, LM-MEL-64 and LM-MEL-70) via linear discriminant analysis (LDA) of SERS signals. a. LDA based on only MCSP biomarker, b. LDA based on MCSP and MCAM biomarkers, c. LDA based on four biomarkers (MCSP, MCAM, ErbB3, and LNGFR).

Figure R3. Clustering three melanoma cell lines (LM-MEL-33, LM-MEL-64 and LM-MEL-70) in response to drug treatment via LDA of SERS signals.

Figure R4. Clustering CTC signatures (Patient 1) in response to drug therapy via LDA of SERS signals.

a

LM-MEL-33

LM-MEL-64

LM-MEL-70

Before drug treatment (Day 0):

(a) LM-MEL-33

(b) LM-MEL-64

(c) LM-MEL-70

On drug treatment (Day 3):

(d) LM-MEL-33

(e) LM-MEL-64

(f) LM-MEL-70

On drug treatment (Day 35):

(g) LM-MEL-33

(h) LM-MEL-64

(i) LM-MEL-70

On drug treatment (Day 70):

(j) LM-MEL-33

(k) LM-MEL-64

(l) LM-MEL-70

Figure R6. SERS intensity distribution for LM-MEL-33, LM-MEL-64, and LM-MEL-70 cell line cells before drug treatment (a-c) and on drug treatment (Day 3, d-f; Day 35, g-i; Day 70, j-l).

Before drug treatment (Day 0):

(a) LM-MEL-33

(b) LM-MEL-64

(c) LM-MEL-70

On drug treatment (Day 3):

(d) LM-MEL-33

(e) LM-MEL-64

(f) LM-MEL-70

On drug treatment (Day 35):

(g) LM-MEL-33

(h) LM-MEL-64

(i) LM-MEL-70

On drug treatment (Day 70):

(j) LM-MEL-33

(k) LM-MEL-64

(l) LM-MEL-70

Figure R7. FACS results for LM-MEL-33, LM-MEL-64, and LM-MEL-70 cell line cells before drug treatment (a-c) and on drug treatment (Day 3, d-f; Day 35, g-i; Day 70, j-l).

Reviewers' comments:

Reviewer #1 (Remarks to the Author):

The authors have adequately addressed the concerns raised in the original review. This has significantly improved the content and potential impact of the study. No further revision is necessary.

Reviewer #2 (Remarks to the Author):

The manuscript entitled "Characterising the Phenotypic Evolution of Circulating Tumour Cells during Treatment" by Tsao et al. describes a novel method for the detection and analysis of melanoma CTCs, based on the detection of labelled antibody binding through Raman spectroscopy. The methodology is of interest given the challenge associated with CTC detection and the need for multiplexing to address tumour/CTCs heterogeneity. However, given the novelty there are many questions regarding sensitivity and specificity that are not fully addressed in the manuscript.

1. Does this technique allow the quantification of cells? The methods do not clearly explain how they obtained the cell images presented relative to the Raman shift plots.
2. What algorithm was used to deconvolute the Raman spectrum of all four Abs into 4 singular sources?
3. For example, based on the profile of the Raman shift (SFig1). How easy is the signal of MCSP-MBA distinguished from LNGFR-MPY?
4. Specificity and sensitivity are only analysed for MCSP- MBA-AuNPs. Considering that patient data is provided with the four antibody combination, more information is required on the sensitivity and specificity of the other three markers, separately and in combination.
5. Can the authors clarify: "A total of 150 Raman 115 measurements are acquired from each sample for analysis". Is that 150 partitions of the sample after staining? This is vital information as underlies the freq vs intensity plots. In flow that is cell by cell usually 10-100 events are counted, in microscopy is usually fields or single cells, but it is hard to relate what are the 115 measurements in here.
6. In page 6 – assay sensitivity. The authors attribute the low intensity observed to the loss of melanoma cells during density gradient and CD45 depletion. However, the authors still detected a signal with the 10 cells spike, but the signal with 250-1000 cell spike was significantly lower suggesting that cells were lost more readily in those groups? Can the signal be related to the number of cells (CTCs) in the fraction?

Other comments:

7. Introduction needs to be more concise. Current manuscript includes results and data on the introduction, plus the last paragraph is better fit to the methods section, rather than introduction.
8. Too many references cite reviews on the subject rather than original work. Just one example: line 99 page 4, low affinity nerve growth factor receptor (LNGFR) 12,24, stem-cell biomarker which is strongly associated with resistance development (34- Clevers, H. The cancer stem cell: premises, promises and challenges. Nat. Med 2011); given that would that LNGFR/CD271 is not a common marker of resistance in melanoma it would be useful to have a more specific reference.
9. In SFig 13, all the labelled indicate LM-MEL-70, but I think it supposed to be 5 different cell lines. Can the authors indicate how many cells were assessed per cell line to obtain that profile?
10. Can the authors indicate whether they found CTCs in all 10 cases analysed? Previous studies in melanoma have found CTCs in 26, 52 and 79% of cases (Khoja 2013, Gray 2015, Luo 2014, respectively). Can the author comment on their high CTC detection rate.

Response to Reviewer's Comments

Reviewer #2:

The manuscript entitled "Characterising the Phenotypic Evolution of Circulating Tumour Cells during Treatment" by Tsao et al. describes a novel method for the detection and analysis of melanoma CTCs, based on the detection of labelled antibody binding through Raman spectroscopy. The methodology is of interest given the challenge associated with CTC detection and the need for multiplexing to address tumour/CTCs heterogeneity. However, given the novelty there are many questions regarding sensitivity and specificity that are not fully addressed in the manuscript.

1. Does this technique allow the quantification of cells? The methods do not clearly explain how they obtained the cell images presented relative to the Raman shift plots.

Reply: We appreciate the Reviewer's recognition of the novelty and importance of this work and the valuable comments. All comments are addressed below.

(1) Our technique is capable of cell quantification as the Raman intensity is proportional to the cell numbers (**Supplementary Figs. 4 and 5**). According to the calibration plot for LM-MEL-64 cells titrated in 10 mL of blood (**Supplementary Fig. 5**), we can quantify cell numbers in an unknown sample based on the collected Raman intensities.

(2) Single-cell SERS image is a false colour image, obtained by using the integrated Raman intensity of the characteristic peak from each antibody-SERS label, typically, Raman peak at 1075 cm^{-1} for MCSP-MBA-AuNPs, 1375 cm^{-1} for MCAM-TFMBA-AuNPs, 1335 cm^{-1} for ErbB3-MNBA-AuNPs, and 1000 cm^{-1} for LNGFR-MPY-AuNPs, respectively. The colour (*i.e.*, blue, red, green and purple) of each spot in **Fig. 2b** represents the distribution of antibody-SERS labels, which further indicates the cell surface marker distribution.

We have explained this in the revised main text (Page 6).

2. What algorithm was used to deconvolute the Raman spectrum of all four Abs into 4 singular sources?

Reply: We performed a Gaussian function to deconvolute the resulting Raman spectrum into separate sources. With this function, we fitted the peaks or separated close peaks according to peak positions, intensities, and full width at half maximum (FWHM).

We have clarified this in the revised Supplementary Information (Page 6).

3. For example, based on the profile of the Raman shift (SFig1). How easy the signal of MCSP-MBA can be distinguished from LNGFR-MPY?

Reply: As the characteristic peaks for MCSP-MBA-AuNPs (1075 cm^{-1}) and LNGFR-MPY-AuNPs (1055 cm^{-1} and 1095 cm^{-1}) have more than 20 cm^{-1} difference, the individual signal could be picked up easily. However, to minimize any potential overlap between peaks and thus obtain more accurate quantitative analysis, we performed the Gaussian function-based spectral deconvolution.

We have clarified this in the revised Supplementary Information (Page 6).

4. Specificity and sensitivity are only analysed for MCSP- MBA-AuNPs. Considering that patient data is provided with the four antibody combination, more information is required on the sensitivity and specificity of the other three markers, separately and in combination.

Reply: We are grateful for these suggestions.

(1) The sensitivity of our technology is relevant to the cell-surface marker expression levels—these vary for different markers, different cell lines and they also vary for patient samples. Typically, using the ErbB3-MNBA-AuNPs probe alone, it is difficult to detect less than 1000 LM-MEL-64 cells due to the low expression of ErbB3 in this cell line (**Figure R1**). As such, given the successful detection of 10 cells from 10 mL of blood using MCSP-MBA-AuNPs alone and the obtainment of CTC phenotypic

profiles in 10 longitudinal patient studies (each with numerous pre- and post-therapy time point analyses) using the combined SERS labels, our technique has sufficient sensitivity for CTC detection and characterisation. Given this, we thus believe that the inclusion of the sensitivity study of other three SERS labels would not be of additional supportive value to our present data. Firstly, the phenotypes of cell lines are different from the melanoma CTCs, which may not reflect the sensitivity in the detection of patient samples. Secondly, many assays based on multi-molecular markers (Hoon DS, *et al.*, *J Clin Oncol* 1995) or –protein markers (J. B. Freeman, *et al.*, *J. Transl. Med.* 2012) have demonstrated a significant sensitivity improvement in patient samples, compared to single marker assays. Thirdly, we have already moved to study significant numbers of patient samples.

We have clarified the sensitivity issue in the revised main text (Page 7).

(2) The specificity of the combined SERS labels has been demonstrated through the MCSP-, MCAM-, ErbB3-, and LNGFR-expressing cell line (LM-MEL-62) and non-expressing cell line (LM-MEL-44), followed by flow cytometry validation (**Fig. 2a and Supplementary Figs. 11-12**). Compared to background signals collected from LM-MEL-44, LM-MEL-62 showed positive SERS signals, which was in line with flow cytometry data.

We have included this explanation in the revised main text (Page 6).

5. Can the authors clarify: “A total of 150 Raman measurements are acquired from each sample for analysis”. Is that 150 partitions of the sample after staining? This is vital information as underlies the freq vs intensity plots. In flow that is cell by cell usually 10-100 events are counted, in microscopy is usually fields or single cells, but it is hard to relate what are the measurements in here.

Reply: Yes, we measured 150 partitions of each sample after staining. In our experimental set-up, each SERS measurement generated one SERS spectrum that was the statistically-averaged result of a large ensemble of labelled cells within the scattering volume. For each sample, 150 measurements were continuously collected to represent different portions of cells that were undergoing Brownian movement in the solution. We then plotted distribution profiles of target marker expression levels over multiple interrogations.

We have clarified this in the revised manuscript (Page 5).

6. In page 6 – assay sensitivity. The authors attribute the low intensity observed to the loss of melanoma cells during density gradient and CD45 depletion. (1) However, the authors still detected a signal with the 10 cells spike, but the signal with 250-1000 cell spike was significantly lower suggesting that cells were lost more readily in those groups? (2) Can the signal be related to the number of cells (CTCs) in the fraction?

Reply: Thanks for your comments.

(1) Yes, reviewer’s opinion is right. As our technology is very sensitive, it is capable of detecting a slight decrease in cell numbers within the scattering volume. As such, the signal with 250-1000 cell spike was significantly lower due to cells were lost more readily in those groups.

We have included this point in the revised main text (Page 7).

(2) The resulting signals are related to the number of cells (CTCs) in the scattering volume.

We have clarified this point in the revised main text (Page 5).

Other comments:

7. Introduction needs to be more concise. Current manuscript includes results and data on the introduction, plus the last paragraph is better fit to the methods section, rather than introduction.

Reply: We appreciate reviewer’s suggestions. We have modified the *Introduction* according to the review’s suggestions in the revised main text (Page 5, working scheme section).

8. Too many references cite reviews on the subject rather than original work. Just one example: line 99 page 4, low affinity nerve growth factor receptor (LNGFR) 12,24, stem-cell biomarker which is strongly associated with resistance development (34- Clevers, H. The cancer stem cell: premises, promises and challenges. *Nat. Med* 2011); given that would that LNGFR/CD271 is not a common marker of resistance in melanoma it would be useful to have a more specific reference.

Reply: Thanks for the suggestions. We have carefully checked other references and cited the original work.

For this example, we have replaced with a new reference (Fallahi-Sichani, M., *et al. Mol. Syst. Biol.* 2017, 13 (1), 905).

9. In SFig 13, all the labelled indicate LM-MEL-70, but I think it supposed to be 5 different cell lines. Can the authors indicate how many cells were assessed per cell line to obtain that profile?

Reply: **Supplementary Fig. 13** shows three melanoma cell lines (LM-MEL-33, 64 and 70). For each cell line, five representative single-cell SERS images (left-hand side) and relative cell-surface marker expression levels of that cell (right-hand side) are indicated. The different expression profiles within the same cell line reflect the cell heterogeneity and reinforce the necessity of comprehensive CTC characterisation.

We have clarified this in the main text (Page 8) and Supplementary Information (Page 19).

10. Can the authors indicate whether they found CTCs in all 10 cases analysed? Previous studies in melanoma have found CTCs in 26, 52 and 79% of cases (Khoja 2013, Gray 2015, Luo 2014, respectively). Can the author comment on their high CTC detection rate.

Reply: We successfully detected CTCs in all 10 cases with active stage IV (metastatic) disease. The high sensitivity of our technology could be attributed to three reasons: (1) free from prior CTC isolation, reducing CTC loss during the isolation process; (2) multi-marker based CTC detection, increasing the probability of detecting rare CTCs; (3) ultra-sensitive and multiplexed detection technology—SERS, being capable of detecting single molecule signals and simultaneously characterising multiple cell-surface markers. In comparison, Khoja, L., *et al. (Ann. Oncol.* 2014) and Luo, X. *et al. (Cell Rep.* 2014) required prior CTC isolation that is antibody-dependent. Such antibody-dependent CTC isolation steps are prone to disregard tumor cells that have low target marker expressions. To improve CTC detection capacity, one potential solution is using multiple markers. Gray, E.S., *et al. (J Invest Dermatol.* 2015) detected multi-marker expressions on the CTCs using multiparametric flow cytometry, whose detection sensitivity is ~2000 to 15,000-fold lower than SERS technology (Han, J. *et al. Biomaterials* 2016). Thus, we believe that our strategy—direct and multiplexed SERS detection has great advantages over others in CTC enumeration and characterisation.

We have included this discussion in the revised main text (Page 10).

Figure R1: Raman signals for different numbers of melanoma cells (LM-MEL-64 conjugated with ErbB3-MNBA-AuNPs) spiked into 1 mL of PBS.

Reviewers' comments:

Reviewer #3 (Remarks to the Author):

The authors addressed most of the comments highlighted in the previous review. However, major technical considerations still need clarification. Given the novelty of the technology in this field (not easily reproduced or verified by an independent laboratory), and the at the same time the conflicting reports in the field of CTCs, it is important to clarify any issue that could result in unspecific signal and therefore over estimation of CTC numbers and phenotype. With that in mind the paper lacks clear description of the procedures, controls and thorough analysis of sensitivity and specificity.

Response to comment 4.

I would disagree with the authors's response to query 4. The definition of sensitivity and specificity of each antibody together and in combination is critical. Specially, no reference have been made to specificity. How can the authors discard that there is not unspecific noise into the reporter channel. Although the authors argue that they can clear distinguish the peaks, given the Raman fingerprints exemplified in figure S1, low peaks on one reporter may be occluded by another reporter. Figure R1, is incomprehensible. The RAMAN signal does not increase with the increase in cells. If the cells were expressing, even low amounts of ERB3, there would be an increase. The results rather will be interpreted as there is no ERB3 expression on the cells and that is just background noise?

I agree that cell lines are not the best representation. However it is problematic that although images are presented, the sample analysis is done in read out over a pool of cells. It is unclear for the results whether there are more cells expressing the different markers of there is an increase in marker expression on a similar number CTCs. That is a limitation of the technology compared to microscopy (most common read out for CTCs).

Finally, the fact that many patients have been analysed using the combination without checking their sensitivity and specificity is not an appropriate justification. Without that information, how can the patient data be critically analysed?

Response to comment 10, line 270-272 in current manuscript:

How can the authors claim that SERS can detect single molecule signal, when they require 10 cells for MCSP detection or 1000 cells of a cell lines with low ERB3 expression?

Supplementary Fig. 13 – still labelled LM-MEL-70 on the top.

Responses to the Reviewer 3#:

The authors addressed most of the comments highlighted in the previous review. However, major technical considerations still need clarification. Given the novelty of the technology in this field (not easily reproduced or verified by an independent laboratory), and the at the same time the conflicting reports in the field of CTCs, it is important to clarify any issue that could result in unspecific signal and therefore over estimation of CTC numbers and phenotype. With that in mind the paper lacks clear description of the procedures, controls and thorough analysis of sensitivity and specificity.

We thank the reviewer for his/her valuable suggestions and comments. We have now clarified details in procedures (**Supplementary Information**, Pages 4-6), and included new data for the sensitivity and specificity (**main text**, Pages 6-7; **Supplementary Information**, Pages 10-13, and 16).

1. I would disagree with the authors's response to query 4. The definition of sensitivity and specificity of each antibody together and in combination is critical. Specially, no reference have been made to specificity. How can the authors discard that there is not unspecific noise into the reporter channel.

Reply: We appreciate the reviewer's suggestions on the demonstration of specificity and sensitivity of each antibody alone and in combination.

(a) Specificity study: Accordingly, we performed experiments to test the specificity of each antibody alone and in combination by using reported cell lines with high marker expression as positive controls and ones with no/low marker expression as negative controls. For example, SK-MEL-28, which has been reported for high expression of MCSP and MCAM (*J. Invest. Dermatol.* 2015, 135, 2040), was chosen as the marker-positive cell line for the specificity study of MCSP and MCAM antibodies. MCF7, which has low expression of MCSP and MCAM, was used as the marker-negative cell line (*J Natl Cancer Inst.* 2010, 102, 1496; *Breast Cancer Res.* 2009, 11, R1; and *PLoS ONE*, 2012, 7, e43752). The detailed information for all marker expression in tested cell lines has been summarized in **Table R1**. To minimize data variations caused by different cell numbers, the same amount of cells from different cell lines were tested across the specificity assay. As shown in **Figure R1 (a-b)**, high SERS signals for MCSP-MBA and MCAM-TFMBA from SK-MEL-28 cells indicated high expression of MCSP and MCAM in SK-MEL-28 cells, while low SERS signals from MCF7 cells indicated low expression of MCSP and MCAM in MCF7 cells, which were consistent with literature reports (*J. Invest. Dermatol.* 2015, 135, 2040; *J Natl Cancer Inst.* 2010, 102, 1496; *Breast Cancer Res.* 2009, 11, R1; and *PLoS ONE*, 2012, 7, e43752). This result demonstrated the specificity of our SERS technique for the detection of MCSP and MCAM individually.

Moreover, we tested the specificity of ErbB3 and LNGFR in MCF7 and SKBR3, and in SK-MEL-28 and bone marrow mesenchymal stem cells (BM-MSC), respectively. Different marker expression levels in these reported cell lines were successfully identified from resulting ErbB3-MNBA and LNGFR-MPY signals (**Figure R1 (c-d)**), respectively, which were in line with literature reports (*Oncotarget.* 2015, 6, 3932; *Oncogene*, 1999, 18, 6050; *J. Invest. Dermatol.* 2015, 135, 2040; *Transfus Med Hemother.* 2008, 35, 168; and *J. Tissue. Eng. Regen. Med.* 2007, 1, 296). Furthermore, we tested the specificity for four antibodies together in both SK-MEL-28 and MCF7 cell lines (**Figure R1 (e)**), which showed consistent results as determined by individual antibodies ($R^2 = 0.996$ for SK-MEL-28, and $R^2 = 0.985$ for MCF7, as indicated in **Figure R1 (f)**). To further demonstrate the specificity of our assay, flow cytometry detection was performed for validation and showed agreement with SERS data (**Figure R2**).

Therefore, given the specificity of each antibody alone and in combination tested in these reported cell lines, we demonstrated that cell surface markers could be specifically identified by SERS technology with neglectable background noise.

We have included the new data and discussion in the revised manuscript (Pages 6-7).

Table R1. Summary of references documenting MCSP, MCAM, ErbB3, and LNGFR expression in different cell lines and our corresponding flow cytometry data for validation.

		Marker expression levels	
		Literatures	Experimental sMFI [§]
MCSP	SK-MEL-28	+++* (J. Invest. Dermatol. 2015, 135, 2040.)	17.77
	MCF7	+ (J Natl Cancer Inst, 2010, 102, 1496.)	1.01
MCAM	SK-MEL-28	+++ (J. Invest. Dermatol. 2015, 135, 2040.)	45.92
	MCF7	- (Breast Cancer Res. 2009, 11, R1; PLoS ONE, 2012, 7, e43752.)	0.98
ErbB3	MCF7	++ (Oncotarget. 2015, 6, 3932; Oncogene, 1999, 18, 6050)	1.53
	SKBR3	+ (Oncotarget. 2015, 6, 3932; Oncogene, 1999, 18, 6050)	1.41
LNGFR	SK-MEL-28	++ (J. Invest. Dermatol. 2015, 135, 2040.)	2.36
	BM-MSC	- (Transfus Med Hemother. 2008, 35, 168. J. Tissue. Eng. Regen. Med, 2007, 1, 296.)	1.22

*Score of reactivity of cells with antibodies: -, <5%; +, 6-20%; ++, 21-60%; and +++, 61-100% (*PLoS ONE*. 2014, 9, e84417).

§The specific MFI represents the ratio of the mean fluorescence intensity for the targeting marker over the mean fluorescence intensity of the isotype control (one representative experiment).

The raw flow cytometry data are shown in Figure R2.

Figure R1. Assay specificity studies of each antibody-SERS label alone and in combination in reported cell lines. a. MCSP expression in SK-MEL-28 and MCF7 cell lines; b. MCAM in SK-MEL-28 and MCF7 cell lines; c. ErbB3 in MCF7, SKBR3, and SK-MEL-28 cell lines; d. LNGFR in SK-MEL-28, BM-MSC, and MCF7 cell lines; e. four target combination in SK-MEL-28 and MCF7 cell lines. f. Comparison between each antibody-SERS label alone and in combination for profiling four cell surface markers in two cell lines. Error bar represents standard deviation with $n = 3$.

Figure R2. Flow cytometry results for the different marker expression in different cell lines.

(b) Sensitivity study: To explore the sensitivity of each antibody-SERS label alone and in combination for the detection of target cells, SK-MEL-28 cell line was used to test the sensitivity of MCSP-, MCAM-, LNGFR- and four target combination-SERS labels, while MCF7 was used for ErbB3-SERS label test. As indicated in Figure R3, each antibody-SERS label alone and in combination enable the detection down to 10 cells, demonstrating that our technique is sensitive for CTC characterisation. The detection sensitivity might vary for different markers, cell lines, and patient samples. However, given that our technique characterised CTCs without an isolation step (minimising CTC loss) and utilized a sensitive SERS technology as the signal read-out, we thus believe this technique could be capable of monitoring CTC phenotype changes during the disease progression and their response to the therapy.

We have included the data and discussion in the revised manuscript (Page 7).

Figure R3. Assay sensitivity study using different numbers of cells (conjugated with each antibody-SERS label alone and in combination) spiked into 1 mL of PBS. a. SK-MEL-28 labelled with MCSP-MBA-AuNPs; b. SK-MEL-28 labelled with MCAM-TFMBA-AuNPs; c. MCF7 labelled with ErbB3-MNBA-AuNPs; d. SK-MEL-28 labelled with LNGFR-MPY-AuNPs; e. SK-MEL-28 labelled with four target combination-SERS labels. Error bar represents standard deviation with n = 3.

2. Although the authors argue that they can clear distinguish the peaks, given the Raman fingerprints exemplified in figure S1, low peaks on one reporter may be occluded by another reporter.

Reply: We agree that low peaks on one reporter may be occluded by another reporter. To minimize this potential spectral overlap, in our study, only peaks with similar intensities were used as reporter characteristic peaks to represent each reporter.

Additionally, we also applied spectral deconvolution based on a Gaussian function, using Fityk 0.9.8 program (*J. Phys. Chem.* 1987, 91, 634, and *J. Appl. Cryst.* 2010, 43, 1126). With this function and program, we fitted peaks or separated close peaks according to peak positions, intensities, and full width at half maximum (FWHM). Comparing SERS spectra of individual MCSP-MBA and LNGFR-MPY to their respective deconvoluted models, we demonstrated that peak intensities at 1075 cm^{-1} and 1000 cm^{-1} in deconvoluted models only have 0% and 1.8% variations to their corresponding experimental spectral data, respectively (**Figure R4 (a-b) and Table R2**). Such high consistence indicates that Gaussian function can reliably deconvolute potential overlapping peaks. Moreover, spectra generated from the mixture of MCSP-MBA and LNGFR-MPY, mixed in volume ratios of 1:1 and 1:5, respectively, were shown in **Figure R4 (c-d)**, respectively. The successful deconvolution of MCSP-MBA from LNGFR-MPY was then performed and indicated in **Figure R4 (c-d)**, in which peaks belonging to MCSP-MBA and LNGFR-MPY can be clearly distinguished from each other. The corresponding peak intensities of MCSP-MBA source in both sets of mixture (**Figure R4 (c-d)**), calculated from deconvoluted models, are in good agreement with respective peak intensities at individual MCSP-MBA levels (**Table R2**), thus demonstrating that deconvolution is capable of resolving the spectral overlap issue.

We have added this discussion in the revised manuscript (Page 5) and **Supplementary Information** (Pages 6, and 8-9).

Figure R4. SERS spectra of (a) MCSP-MBA-AuNPs, (b) LNGFR-MPY-AuNPs, (c) the mixture of MCSP-MBA- and LNGFR-MPY-AuNPs in a volume ratio of 1:1; and (d) the mixture of MCSP-MBA- and LNGFR-MPY-AuNPs in a volume ratio of 1:5. Solid lines are experimental spectra, and dash lines are deconvoluted models.

Table R2. Peak intensities in experimental spectra and deconvoluted models

	Characteristic peaks (cm ⁻¹)	Peak intensity in experimental spectra (a.u.)	Peak intensity in deconvoluted models (a.u.)
MCSP-MBA	1075	850	850
LNGFR-MPY	1000	759	745
MCSP-MBA:LNGFR-MPY = 1:1 v/v %	1075	985	1024 (849 for MBA source)
	1000	573	577 (577 for MPY source)
MCSP-MBA:LNGFR-MPY = 1:5 v/v %	1075	424	392 (165 for MBA source)
	1000	742	728 (728 for MPY source)

3. Figure R1 (in the previous review), is incomprehensible. The RAMAN signal does not increase with the increase in cells. If the cells were expressing, even low amounts of ERBB3, there would be an increase. The results rather will be interpreted as there is no ERBB3 expression on the cells and that is just background noise?

Reply: Thanks for the valuable comment. We reassessed the sensitivity of ErbB3-MNBA-AuNPs more carefully by exactly following the procedure used for all the study and found that the sensitivity could go down to 10 cells (**Figure R5**). The lower sensitivity shown in the previous review might be due to the decreased SERS signal of ErbB3-MNBA-AuNPs caused by a smaller amount of Raman reporters (MNBA, which was reduced from DTNB by NaBH₄ treatment) being conjugated to the Au surface.

To avoid this confusion, we have now detailed procedures in the **Supplementary Information** (Page 5).

Additionally, we further investigated the sensitivity of ErbB3-MNBA-AuNPs in a reported cell line (i.e., MCF7), as shown in **Figure R3 (c)**, and as low as 10 cells can be detected. Given the successful detection of a relatively low-expressing marker using 10 cells from either of two different cell lines, we believe ErbB3- and other antibody-SERS labels could satisfy the requirement of CTC phenotype characterisation.

We have now included the sensitivity result for ErbB3 in the revised manuscript (Page 7).

Figure R5. Raman signals for different numbers of melanoma cells (LM-MEL-64 conjugated with ErbB3-MNBA-AuNPs) spiked in 1 mL of PBS. Error bars represent standard deviation with n = 3.

4. I agree that cell lines are not the best representation. However it is problematic that although images are presented, the sample analysis is done in read out over a pool of cells. It is unclear for the results whether there are more cells expressing the different markers or there is an increase in marker expression on a similar number CTCs. That is a limitation of the technology compared to microscopy (most common read out for CTCs).

Reply: We agree that microscopy (the most common read out for CTCs) could answer ‘whether there are more cells expressing different molecules or there is an increase in marker expression on a similar number CTCs’. Nevertheless, imaging requires cell isolation, which is a big hurdle and often confounds results due to the CTC loss during the isolation process. The beauty of our technology is that we do not need to isolate cells (even though this can be done if necessary). Additionally, we can easily measure the molecular make-up of CTCs through simultaneously tagging cells for imaging or the bulk assay. Our bulk assay is complementary to microscopy and enables the characterisation of CTC heterogeneity with a simpler and faster workflow. As for using the bulk assay for CTC enumeration, if we do 150 measurements over a pool of cells, we believe we could see more often a “positive” signal in a channel if there are more CTCs in the same amount of blood. The bulk assay is not ideal for CTC counting due to a range of confounding factors, but could be a big step towards

understanding CTC characteristics (i.e. phenotypes) and promoting CTC clinical applications.

We have now included the following sentence in the revised main text (Page 12): “It is also worth noting that the observed signal changes may arise from different amounts of cells expressing target markers or from marker expression changes on a similar number CTCs. Nonetheless, given that our technique is capable of effectively evaluating CTC phenotypes and heterogeneity in response to therapy, we thus believe it could be a big step towards understanding CTC characteristics (i.e. phenotypes) and promoting CTC clinical applications.”

5. Finally, the fact that many patients have been analysed using the combination without checking their sensitivity and specificity is not an appropriate justification. Without that information, how can the patient data be critically analysed?

Reply: Thanks for the comments. We have now added the new data for the sensitivity and specificity study on each antibody-SERS label alone and in combination (Figures R1 and R3), which indicates that our technique can sensitively and specifically detect and analyse CTCs. New figures and discussion have been included in the revised manuscript (Pages 6-7) and Supplementary Information (Pages 10-13, and 16).

6. Response to comment 10, line 270-272 in current manuscript:

How can the authors claim that SERS can detect single molecule signal, when they require 10 cells for MCSP detection or 1000 cells of a cell lines with low ERB3 expression?

Reply: Thanks for the comments. The statement that SERS can detect single molecule signals was to indicate the great potential of SERS in achieving single molecule sensitivity as previously reported (*Science* 1997, 275, 1102 and *Phys. Rev. Lett.* 1997, 78, 1667.), which requires special conditions such as forming ‘hot spots’ within nanoparticles (*ACS Nano* 2016, 10, 7323, and *ACS Nano* 2017, DOI: 10.1021/acsnano.7b00531.). Furthermore, as the aim of our study is to use SERS for clinical applications, we need to achieve a balance among detection factors such as sensitivity, reproducibility, and stability. To this end, we synthesised stable gold nanoparticles as SERS labels for CTC characterisation. Our SERS labels require 10 cells for reliable MCSP expression detection, and we believe that this detection sensitivity is relevant for clinical CTC detection requirements (*ACS Sens.* 2017, 2, 193).

To make it clearer, we have clarified this statement in the revised manuscript (Pages 4 and 10).

7. Supplementary Fig. 13 – still labelled LM-MEL-70 on the top.

Reply: Thanks for the suggestion. We have reorganized the figure by combining two tables in separate pages (in the previous review) into one table, in which three columns correspond to three cell lines (LM-MEL-33, 64, and 70), respectively (**Supplementary Information Fig. 17**).

Supplementary Figure 17: Single-cell SERS images of LM-MEL-33, 64 and 70 cell lines conjugated with 4 Ab-SERS labels. Five single cells were randomly selected and imaged from each cell line. The y-axis (percentage) in the histogram is calculated by counting the numbers of dots that generated positive signals on the individual cell (i.e., Percentage = the pixel numbers of dots/the pixel numbers of per cell). Blue, red, orange, and green bars represent MCSP, ErbB3, LNGFR, and MCAM, respectively.

REVIEWERS' COMMENTS:

Reviewer #3 (Remarks to the Author):

The authors have addressed all my previous comments and extensively modified the manuscript as a results.

I believe the manuscript is publishable, however, I am still unclear how easily this method would be implemented or used by others to corroborate the data.

Response to the Reviewer #3:

The authors have addressed all my previous comments and extensively modified the manuscript as a results.

I believe the manuscript is publishable, however, I am still unclear how easily this method would be implemented or used by others to corroborate the data.

Reply: We thank for the reviewer's valuable comments that help us further improve the quality of our paper. To ensure that this method could be implemented by others, we have now prepared the step-by-step protocol used in this manuscript. We also recruited three volunteers, having a background for either nanoparticle preparation or biological experiments, to use our method for tumor cell characterisation. We found that all of them independently completed experiments by exactly following the protocol. According to their feedback and performance, we have now further clarified and detailed our protocol and the **Method** section (main text, Pages 14-17).

Step-by-step protocols

1. Antibody-SERS label preparation.

1.1. Gold nanoparticle synthesis.

- (a) Heat 100 mL of HAuCl_4 (10⁻²% by weight) to boiling.
- (b) Add 0.7 mL of Na_3 -citrate (1% by weight) into the boiling HAuCl_4 solution, and continuously boil for 20 min. The mixture was stirred at 900 rpm during the whole process.
- (c) Cool down the Au colloid solution to room temperature (RT) and keep at 4 °C for storage.

1.2. AuNPs functionalizing with antibodies (Abs) and Raman reporters.

- (a) Add 10 μL of 1 mM Raman reporters (MBA, MNBA, MPY, or TFMBA) and 2 μL of 1 mM MUA (antibody conjugation linker) in ethanol into 1 mL of AuNP suspension. To prepare MNBA reporters, fresh 300 μL of 20 mM NaBH_4 was added into 2 mL of 5 mM DTNB. To prevent the AuNP aggregation during MPY reporters being functionalized to the AuNP surface, 20 μL of 0.1 M NaOH was added into the AuNP suspension to adjust pH to 10.
- (b) Incubate the mixture at RT for 5 h. The mixture was shaken at 350 rpm during the incubation.
- (c) Centrifuge the mixture at 7600 rpm for 10 min, discard the supernatant, and resuspend the concentrated Raman reporter-conjugated AuNPs (called as 'SERS labels') into 200 μL of HEPES buffer (pH = 5.9).
- (d) Add 40 μL of 3.33 mg mL^{-1} EDC and 40 μL of 2 mg mL^{-1} Sulfo-NHS in HEPES buffer into the mixture.
- (e) Incubate the mixture at RT for 20 min. The mixture was shaken at 350 rpm during the incubation.
- (f) Centrifuge the mixture at 7600 rpm for 10 min, and then discard the supernatant.
- (g) Resuspend the concentrated SERS labels into 200 μL of 0.1 mM PBS.
- (h) Add 1 μg of either human anti-MCSP, anti-MCAM, anti-ErbB3, and anti-LNGFR mouse monoclonal antibodies or isotype-matched IgG into the SERS label solution.
- (i) Incubate the mixture for 0.5 h at RT. The mixture was shaken at 350 rpm during the incubation.
- (j) Centrifuge Ab-SERS labels at 600 \times g at 4 °C for 8 min, and then discard the supernatant.
- (k) Resuspend the concentrated Ab-SERS labels in 200 μL of 0.1% BSA for 0.5 h at RT and keep at 4 °C for storage.

2. Clinical sample treatment

2.1. PBMC isolation. PBMC isolation was performed using Ficoll-Paque PLUS (GE Healthcare Life Science) according to manufacturer's protocol.

- (a) Collect 10 mL of blood samples in EDTA containing 50 mL falcon tubes, and process within 4 hours from the collection over Ficoll-Paque PLUS (GE Healthcare Life Science).
- (b) Dilute the blood sample with 10 mL of a balanced salt solution consisting of 1 volume of solution A (5.5 mM Anhydrous D-glucose, 5 mM $\text{CaCl}_2 \cdot 2\text{H}_2\text{O}$, 0.98 mM $\text{MgCl}_2 \cdot 6\text{H}_2\text{O}$, 5.4 mM KCl, and 145 mM TRIS) and 9 volumes of solution B (140 mM NaCl).
- (c) Add 15 mL of Ficoll-Paque PLUS to the centrifuge tube.
- (d) Carefully layer the 20 mL of diluted blood sample onto the Ficoll-Paque PLUS.
- (e) Centrifuge at $400 \times g$ for 30 min at 20°C .
- (f) Draw off the upper layer using a clean Pasteur pipette, leaving the lymphocyte layer undisturbed at the interface. Care should be taken not to disturb the lymphocyte layer.
- (g) Transfer the lymphocyte layer to a clean centrifuge tube.
- (h) Add 3 volumes of balanced salt solution to the lymphocytes in the test-tube (e.g., for 2 mL of lymphocyte samples, add 6 mL of balanced salt solution).
- (i) Suspend the cells by gently drawing them in and out of a Pasteur pipette.
- (j) Centrifuge at $100 \times g$ for 10 min at 20°C and discard the supernatant.
- (k) Suspend the lymphocytes in 6 mL of a balanced salt solution by gently drawing them in and out of a Pasteur pipette.
- (l) Centrifuge at $100 \times g$ for 10 min at 20°C , and discard the supernatant.
- (m) Isolated PBMCs from each 10 mL of blood were stored in a CryoTubes (Corning) containing 80% RF10, 10% DMSO and 10% FCS at -80°C .

2.2. CD45 depletion. PBMCs were depleted with the EasySep Human CD45 depletion kit (StemCell) according to manufacturer's protocol.

- (a) Prepare PBMC suspension at a concentration of 1×10^7 - 5×10^7 in 2 mL of recommended medium (PBS containing 2% FCS and 1 mM EDTA). Cells must be placed in a 5 mL (12×75 mm) polystyrene tube to fit into the EasySep® Magnet.
- (b) Add 50 μL of EasySep® CD45 Depletion Cocktail into the sample mixture. Mix well and incubate at RT for 15 min.
- (c) Add 100 μL of EasySep® Magnetic Nanoparticles into the sample mixture. Mix well and incubate at RT for 10 min.
- (d) Bring the cell suspension to a total volume of 2.5 mL by adding the recommended medium. Mix the cells in the tube by gently pipetting up and down 2-3 times. Place the tube (without cap) into the magnet. Set aside for 10 min.
- (e) Pick up the EasySep® Magnet, and in one continuous motion invert the magnet and tube, pouring off the supernatant fraction into a new tube. The magnetically labelled cells will remain inside the tube, held by the magnetic field of the magnet. Leave the magnet and tube in inverted position for 3 sec, then return to upright position.

(f) Remove the original tube from the magnet and place the new tube containing the decanted supernatant into the magnet. Set aside for 10 min.

(g) Repeat Step f. The CD45-depleted cells in the new tube are then ready for use.

3. Ab-SERS labelling for CTC detection.

(a) Collected cells suspended in 200 μL of FPBS buffer (PBS containing 1% FCS) were incubated with the mixture of four Ab-SERS labels (30 μL each) at 37 $^{\circ}\text{C}$ for 30 min. Before applying for cell labelling, Ab-SERS labels were centrifuged at 400 \times g for 2 min, and the Ab-SERS label supernatants were then used for applications.

(b) Centrifuge the mixture at 400 \times g for 1 min, and then discard the supernatant.

(c) Resuspend the mixture in 200 μL of FPBS buffer, and repeat the step b three times.

(d) Resuspend the samples in 60 μL of FPBS buffer, and place into a cuvette for SERS measurements.

4. SERS measurements.

4.1. Bulk assay. SERS spectra were recorded with a portable IM-52 Raman Microscope (Snowy Range Instruments). The 785 nm laser wavelength was used for excitation of Raman scattering. SERS spectra were obtained at 1 sec integration time with a laser power of 70 mW.

(a) Continuously measure 50 SERS spectra from each sample, and then stop to resuspend cells.

(b) Repeat step a twice. As such, 150 SERS spectra were generated from each sample.

4.2. Single-cell SERS imaging. SERS images were recorded with the Witec alpha 300R microscope with 632.8-nm line from a HeNe laser as excitation and obtained at 100 ms integration time with an EMCCD, using a 20 \times microcopy objective.

(a) Cell suspension was deposited onto glass slides using cytocentrifuge for single-cell SERS imaging.